# C-type natriuretic peptide attenuates enhanced glycolysis and de novo pyrimidine synthesis in pericytes of patients with pulmonary arterial hypertension
Minhee Noh[1], Ankita Mitra[2], Lisa Krebes[1], Werner Schmitz[3], Jan Dudek[4], Stuti Agarwal[2], Christoph Maack [4], Paula Arias-Loza[5], Takahiro Higuchi [5], Ivan Aleksic [6], Vinicio A. de Jesus Perez [2], Michaela Kuhn[1] & Swati Dabral [1] ✉

Metabolic reprogramming of vascular cells plays a crucial role in Pulmonary Arterial Hypertension (PAH), marked by a shift from oxidative phosphorylation to glycolysis (Warburg effect), altered purine biosynthesis, impaired glutaminolysis and fatty acid oxidation, driving endothelial and smooth muscle cell hyperproliferation. The metabolic alterations underlying pericyte dysfunction in PAH remain largely unexplored. Here, we investigated the metabolic alterations in PAH lung pericytes and the impact of C-type natriuretic peptide (CNP) and Guanylyl Cyclase-B/cyclic GMP signaling on these changes. Our results demonstrate that PAH pericytes exhibit increased glucose uptake, glycolysis, and de novo pyrimidine synthesis, promoting their hyperproliferation. These changes are driven by the upregulated glucose transporter, GLUT-1 and Pyruvate dehydrogenase kinase 1, along with enhanced CAD (Carbamoyl-phosphate synthetase 2, Aspartate transcarbamoylase, and Dihydroorotase) activity, both in vitro and in situ. CNP counteracts these alterations through activation of cGMP-dependent kinase I, reducing HIF-1α and GLUT-1 expression and thereby glucose uptake. Additionally, CNP activates Phosphodiesterase 2 A and thereby inhibits CAD activation and de novo pyrimidine synthesis. Accordingly, CNP prevented growth factor-induced proliferation and metabolic changes in murine pericytes within precision-cut lung slices. This study highlights dysregulated metabolic pathways in PAH pericytes and the therapeutic potential of CNP.

Metabolic reprogramming of vascular cells contributes to vascular remodeling and dysfunction in Pulmonary Arterial Hypertension (PAH). Alterations in the cellular metabolic machinery satisfy the increased energy demands of the hyperproliferating endothelial and smooth muscle cells[1,2]. A major mechanism in PAH is the shift from oxidative phosphorylation to cytoplasmic glycolysis, known as the Warburg effect[3]. Stabilization of the transcription factor Hypoxia inducible factor-1α (HIF-1α) drives augmented expression of the glucose transporter GLUT-1 and of several glycolytic enzymes[4]. Other hallmarks are increased de novo purine synthesis and impaired glutaminolysis and fatty acid oxidation[5–7]. Such metabolic alterations not only sustain but also further enhance the hyperproliferation and migration of pulmonary arterial and arteriolar endothelial and smooth

[1]Institute of Physiology, University of Würzburg, Würzburg, Germany. [2]Divisions of Pulmonary and Critical Care Medicine and Stanford Cardiovascular Institute, Stanford University, California, CA, USA. [3]Institute of Biochemistry and Molecular Biology, University of Würzburg, Würzburg, Germany. [4]Comprehensive Heart Failure Center, University Hospital Würzburg, Würzburg, Germany. [5]Department of Nuclear Medicine and Comprehensive Heart Failure Center, University Hospital Würzburg, Würzburg, Germany. [6]Department of Thoracic and Cardiovascular Surgery, University Hospital Würzburg, Würzburg, Germany. ✉e-mail: swati.dabral@uni-wuerzburg.de

muscle cells as well as of adventitial fibroblasts, thereby promoting macrovascular thickening[8–10]. Recent studies revealed that these metabolic and functional alterations extend to microvascular pericytes from PAH patients[11]. However, the specific metabolic pathways contributing to pericyte hyperproliferation are only partly known. Deciphering the metabolic alterations contributing to lung pericyte dysfunction in PAH patients will increase our understanding of the pathogenesis of this disease and may unravel targets for novel therapeutic interventions.

The paracrine acting endothelial hormone C-type natriuretic peptide (CNP) plays a crucial role in cardiovascular and metabolic homeostasis. Through its transmembrane guanylyl cyclase-B (GC-B) receptor, forming cyclic GMP as second messenger, CNP "relaxes" pericytes and thereby lowers peripheral microvascular resistance, strengthens endothelial barrier functions, moderates the activity of leukocytes and platelets, and prevents atherogenesis[12–14]. Moreover, CNP improves glucose utilization and lipid metabolism in adipocytes[15]. The role(s) of CNP in the pulmonary vasculature are less explored. Notably, our recent work showed that the CNP/GC-B/cGMP signaling pathway is preserved in cultured PAH pericytes and prevents Platelet derived growth factor-BB (PDGF-BB) - induced proliferation and migration. This effect is partly mediated by inhibition of PDGF-BB - induced PI3K/AKT mediated FoxO3 degradation, which stabilizes the nuclear activity of this transcription factor[16]. Interestingly, CNP mRNA levels are significantly reduced in the lungs of PAH patients and PH animals, suggesting a potential pathophysiological relevance of this pathway. Whether CNP/GC-B signaling also regulates cellular, e.g. pericyte metabolism is unknown.

Here we performed extensive metabolomic studies of cultured healthy control and PAH lung pericytes. The results demonstrate that PAH pericytes exhibit an altered metabolic phenotype with augmented glucose uptake, increased glycolysis, and de novo pyrimidine synthesis, all contributing to their increased intrinsic proliferative capacity. These changes are driven by the upregulation of glucose transporter, GLUT-1, and Pyruvate dehydrogenase kinase 1, and increased activity of multifunctional enzyme CAD (Carbamoyl-phosphate synthetase 2, Aspartate transcarbamoylase, and Dihydroorotase) in PAH pericytes, both in vitro and in situ. Notably, CNP inhibited these metabolic and functional alterations. In both control and PAH pericytes, CNP attenuated glucose uptake via activation of cyclic GMP-dependent kinase I (cGKI) and subsequent inhibition of the expression of HIF-1α and its target GLUT-1. In addition, in the PAH pericytes CNP/GC-B signaling inhibited the expression, phosphorylation and activation of CAD, which catalyzes the rate limiting step of de novo pyrimidine synthesis. This effect was mediated by cGMP-dependent activation of Phosphodiesterase 2 (PDE2A) and a negative cGMP-to-cAMP crosstalk, ultimately resulting in inhibition of MAPK/ERK42/44 activity. Notably, this second signaling pathway of CNP was only observed in PAH pericytes, not in controls. Corroborating these observations in cultured human lung pericytes, CNP prevented PDGF-BB-induced proliferation and metabolic gene expression of murine pericytes in situ, in cultured precision cut lung slices (PCLS). These molecular studies together dissect novel pathways leading to the metabolic dysregulation of diseased, activated human lung pericytes and show their counter regulation by the hormone CNP.

## Results
### Cultured lung pericytes from patients with PAH exhibit a distinct metabolic profile, with increased glucose uptake and glycolysis
As recently described by us[16], lung pericytes isolated from PAH patients exhibit slightly increased basal proliferation rates, which were markedly enhanced by PDGF-BB (Fig. 1a). To obtain insights into the metabolic fate of glucose as the carbon source and it's role in maintaining the hyperproliferative phenotype of PAH pericytes, we combined steady state metabolite profiling with [U-$^{13}$C$_6$] glucose isotope tracing to examine isotopic enrichment. The metabolomes of control and PAH pericytes were compared under baseline conditions and after PDGF- BB stimulation (30 ng/ml PDGF-BB during 24 h; 3 donors per group). Score plots generated by principal component analyses (PCA) revealed a distinct clustering of

metabolites in PAH and control pericytes at baseline (Fig. 1b, left panel) as well as after PDGF-BB stimulation (Fig. 1b, right panel). Figure 1c shows the schematic overview of $^{13}$C$_6$-glucose tracing during glycolysis. Evaluation of the uptake and utilization of $^{13}$C$_6$-glucose showed mildly but significantly increased $^{13}$C$_6$-glucose levels in PAH pericytes at baseline (Fig. 1c, top) and after PDGF-BB treatment (Fig. 1c, lower panel). Concomitantly, $^{13}$C labeling of the proximal glycolytic metabolites Glucose-6-phophate (M + 6), glycerate-3-phospate (M + 3), and phosphoenol pyruvate (M + 3) was increased in PAH pericytes (Fig. 1c). Together these data indicate a higher glycolytic metabolite enrichment in PAH pericytes.

To validate the isotope tracing findings, we compared glucose uptake and glycolytic activity of control and PAH pericytes by measuring 2′-deoxy-2′-[$^{18}$F]fluoro-D-glucose ($^{18}$F-FDG) uptake and extracellular acidification rates (ECAR, by seahorse analyzer). PDGF-BB (30 ng/ml,24 h) stimulated glucose uptake in control and, more, in PAH pericytes (Fig. 1d). Moreover, PAH pericytes had higher ECAR at baseline and after PDGF-BB treatment (Fig. 1e). In line with these results, the expression of GLUT-1, the most abundant glucose transporter in pericytes[17], was increased in PAH pericytes (Fig. 1f: immunoblots with whole cell proteins; Supplementary Fig. 1a: immunoblots with enriched cell membrane proteins).

Pointing to the in vivo relevance of these in vitro data, immunohistochemical studies of lung tissue sections demonstrated increased expression of GLUT-1 in specimens from PAH patients as compared to controls (4 lungs per group were studied) (Fig. 1g and Supplementary Fig. 1b). PDGF receptor (PDGFR) -β and Ulex europaeus I agglutinin (ULEX) were used as pericyte and endothelial markers, respectively. Co-staining of PDGFR-β revealed upregulation of GLUT-1 in pericytes in situ (Fig. 1g and Supplementary Fig. 1b depict sections from four lungs per group; yellow arrows in merged pictures point to pericytes). To elucidate whether GLUT-1 participates in PDGF-BB-induced pericyte glucose uptake and proliferation, we tested the effects of the GLUT-1 inhibitor, BAY-876. As shown in Fig. 1h, i, pretreatment with BAY-876 (200 nM, 20 min) significantly attenuated the stimulatory effects of PDGF-BB on $^{18}$F-FDG uptake and proliferation in both control and PAH pericytes.

Together, these data reveal that PAH pericytes have a distinct metabolic profile, with augmented glycolysis. Such alterations are further enhanced by growth factors such as PDGF-BB. Increased GLUT-1 expression participates in this metabolic shift by mediating enhanced glucose uptake, which ultimately fosters pericyte hyperproliferation.

### PAH pericytes exhibit increased de novo pyrimidine synthesis in response to PDGF-BB
To further characterize the distinct metabolic profiles of control and PAH pericytes, we conducted an enrichment analysis of the metabolomic data using MetaboAnalyst 6.0. Quantitative Metabolite set enrichment analyses (MSEA) revealed differences in pathways associated with citric acid cycle, malate-aspartate shuttle and inositol metabolism already under basal conditions (Supplementary Fig. 2a). However, the labeling studies did not reveal significant differences in the incorporation of $^{13}$C$_2$-labeled carbons (M + 2) into intermediates of the TCA cycle in PAH pericytes compared to controls at baseline or under PDGF-BB stimulation (Supplementary Fig. 2b, c). Moreover, the labeling of the TCA cycle components overall was very weak. Together with previous reports[18,19], this indicates a low reliance of lung pericytes on the TCA cycle for energy requirements, even during PDGF-BB-driven hyperproliferation.

Notably, quantitative MSEA revealed that pyrimidine, glutamate, and pentose phosphate metabolism were the metabolic pathways which mostly differed between PDGF-BB-treated PAH and control pericytes (Fig. 2a). Supporting this observation, fold change (FC) analyses revealed that carbamoyl phosphate and phosphoribosyl pyrophosphate (PRPP), both intermediates in de novo pyrimidine synthesis, were strongly upregulated in PAH pericytes compared to controls under baseline conditions as well as after PDGF-BB stimulation (Fig. 2b).

De novo pyrimidine synthesis (DNPyS) is initiated by the trifunctional enzyme complex CAD, consisting of carbamoyl phosphate synthetase,

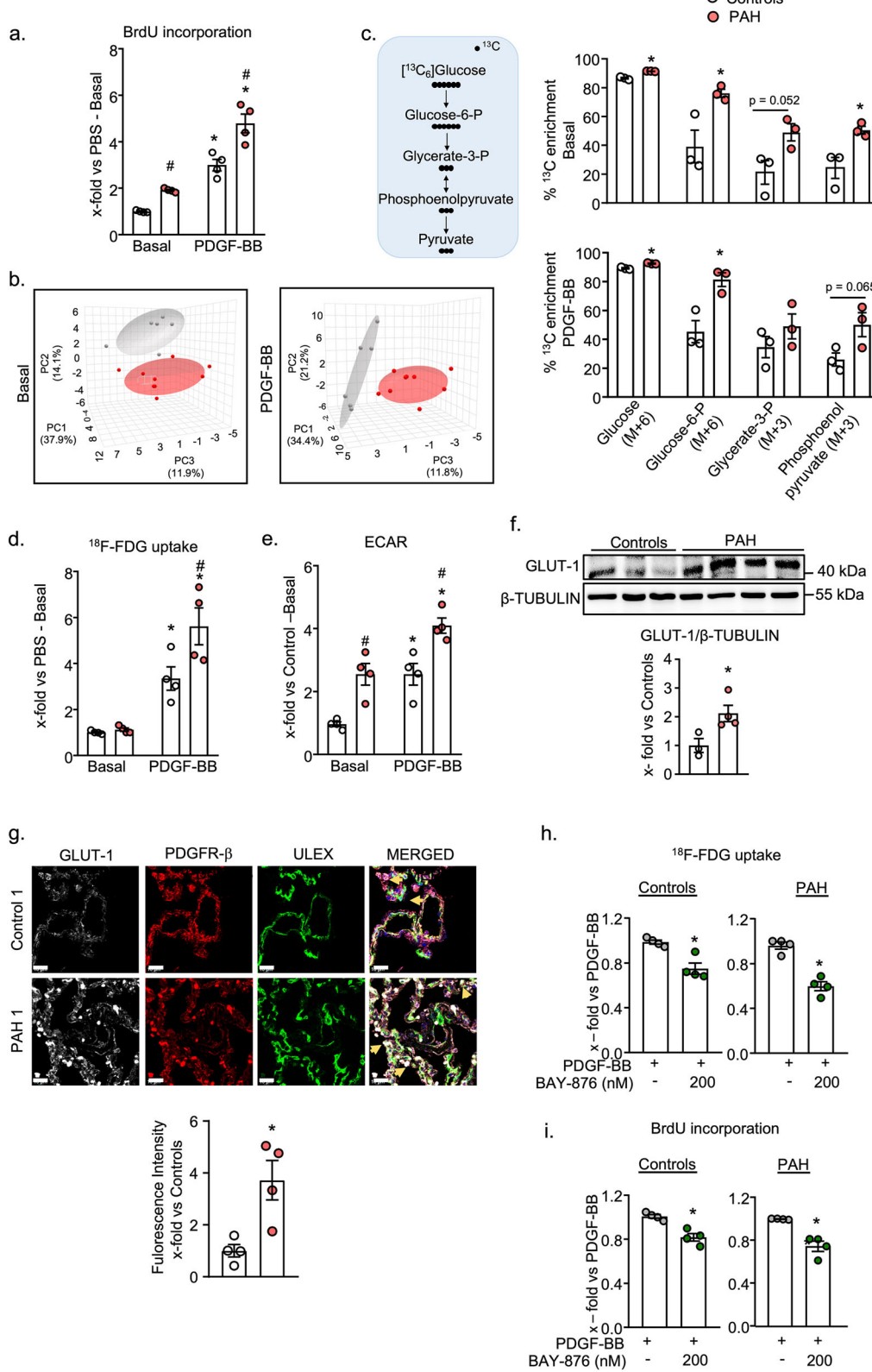

aspartate carbamoyltransferase and dihydroorotase. CAD converts $CO_2$ and glutamine to carbamoyl aspartate, carbamoyl phosphate and dihydroorotic acid sequentially (scheme in Fig. 2c). PRPP, derived from Pentose-5-phosphate, is an allosteric activator of CAD[20]. Next steps are catalyzed by the enzymes dihydroorotate dehydrogenase (DHODH), and uridine monophosphate synthase (UMPS). This synthesis pathway requires the

ribose moieties glutamate and aspartate, resulting in uridine monophosphate (UMP)[21] (Fig. 2c). To follow-up whether DNPyS is enhanced in PDGF-BB-stimulated PAH pericytes (as compared to controls), we examined the labeling patterns of the intermediate and end products of this pathway. There was a trend to increased labeling of Pentose-5-phosphate (M + 5) and its downstream products, uridine (M + 5) and cytidine

**Fig. 1 | Cultured hyperproliferative PAH lung pericytes exhibit distinct metabolic profile, higher glycolytic $^{13}$C labeling and GLUT-1 expression in comparison to control pericytes. a** In comparison to control pericytes, PAH pericytes exhibit higher proliferation rates at baseline and in response to PDGF-BB (30 ng/ml, 24 h) (n = 4 biological replicates for control and PAH; 2 -way ANOVA). The new experiments illustrated in this panel reproduce findings from our earlier study (Dabral S et al. Commun Biol. 2024) for comparison and validation. **b** Score plots based on Principle Component Analysis (PCA) reveal distinct metabolic profiles of control (gray) and PAH (pink) pericytes under (**b**, left panel) baseline and (**b**, right panel) PDGF-BB stimulation on targeted metabolomics analysis. **c** Schematic representation of $^{13}$C labeling patterns after the metabolism of $^{13}C_6$ -glucose through glycolysis (**c**, left panel) and percentage $^{13}$C enrichment in the glycolytic intermediates indicate increased isotope enrichment in PAH pericytes under basal conditions (**c**, upper right panel) and PDGF-BB stimulation (**c**, bottom right panel). Data are shown as percent isotope enrichment, normalized to the total signal (M + 0 to M + n). (for **b** and **c**, n = 3 controls and n = 3 PAH pericytes. **c**: unpaired Student's t test). PAH pericytes had significantly higher glucose uptake (**d**) under PDGF-BB

stimulation (30 ng/ml, 24 h) and increased aerobic glycolysis, Extracellular acidification rate (ECAR) (**e**) under both baseline and PDGF-BB stimulation (**d**: $^{18}$F-FDG glucose uptake assay, n = 4 controls and PAH; **e** BrdU incorporation assay, n = 4 controls and PAH; 2 -way ANOVA). **f** PAH pericytes exhibit higher glucose transporter, GLUT-1 protein expression compared to controls (n = 3 controls and n = 4 PAH patients; unpaired 2-tailed Student's t test). **g** Representative immunofluorescence staining of GLUT-1 on control and IPAH patient lung tissues, followed by fluorescence quantification from n = 4 tissues from each group. GLUT-1 (white), PDGFR-β (red: pericyte marker), ULEX (green: endothelial stain), and nuclei were stained by DAPI (blue). Arrows indicate the colocalization of PDGFR-β with GLUT-1. Scale bar = 20 μm. BAY 876 (200 nM, 30 min) attenuated PDGF-BB (30 ng/ml) induced glucose uptake (**h**), and proliferation (**i**) of control and PAH pericytes (**h**: $^{18}$F-FDG glucose uptake assay, **i**: BrdU incorporation assay, n = 4 controls and 4 PAH; unpaired Student's t test. For **a, d, e**: *p < 0.05 vs PBS-Control, #p < 0.05 vs PDGF-BB-Control. For **c, f** and **g**: *p < 0.05 vs Controls. For **h** and **i** *p < 0.05 vs PDGF-BB. Data is presented as Mean ± SEM.

(M + 5), in PDGF-BB-stimulated PAH pericytes compared to controls, although the changes were not statistically significant (Fig. 2d, upper panel). Notably, PAH pericytes also showed elevated levels of uracil (M + 2) and cytidine (M + 2), while the levels of the corresponding M + 1 isotopologues were unaltered. This enrichment pattern suggests dominant metabolic routing through higher-labeled intermediates (Fig. 2d, lower panel).

The distinct M + 2 labeling pattern of uracil and cytidine is obtained from carbamoyl aspartate which in turn is derived through aspartate (M + 3), a reaction catalyzed by CAD. Aspartate derives from the anaplerotic conversion of pyruvate to oxaloacetate by pyruvate carboxylase (PC)[22]. On the other hand, Pyruvate dehydrogenase kinase 1 (PDK1) inhibits pyruvate dehydrogenase thereby preventing the conversion of pyruvate to Acetyl-CoA[23] and increasing pyruvate availability for pyruvate carboxylase (Scheme in Fig. 2e). To dissect the mechanism(s) augmenting this metabolic pathway in PAH pericytes, we compared the protein expression of the involved enzymes between cultured PAH and control pericytes. CAD expression did not differ between the two groups (Fig. 2f). However, the activating phosphorylation of CAD at Threonine 456 (pCAD$_{Thr456}$)[24] was enhanced in PAH pericytes (Fig. 2f). Pyruvate carboxylase (PC) expression was unchanged, but PDK-1 expression was significantly increased in cultured PAH pericytes (Supplementary Fig. 3a).

Supporting the in vivo relevance of these in vitro data, immunohistochemical studies of tissue sections demonstrated strongly increased levels of pCAD$_{Thr456}$ (Fig. 2g and Supplementary Fig. 4) and PDK1 (Supplementary Fig. 3b) in PDGFR-β$^+$ pericytes within lungs of PAH patients as compared to controls (Fig. 2g, Supplementary Fig. 4 and Supplementary Fig. 3b depict sections from four lungs per group).

Given that PDK1 is upregulated in PAH pericytes, we investigated whether Dicholoracetate (DCA), an inhibitor of PDK affects the metabolic signatures and proliferation of PAH pericytes. As expected, DCA significantly reduced lactate levels, confirming effective inhibition of the Warburg effect (Supplementary Fig. 3c). DCA treatment also led to a modest but statistically significant reduction ( ~ 18%) in PDGF-BB-induced pericyte proliferation (Supplementary Fig. 3d). Interestingly, DCA had no effect on aspartate M + 3 labeling or carbamoyl aspartate levels (Supplementary Fig. 3e, f). Furthermore, the pericyte levels of M + 5- and M + 2-labeled nucleotides were also not changed by DCA, suggesting that DNPyS was not influenced by this compound (Supplementary Fig. 3g). Hence, these experiments reveal an interesting difference in the metabolic effects of CNP and DCA.

To elucidate whether augmented DNPyS participates in PDGF-BB-driven hyperproliferation of PAH pericytes, we tested the effect of siRNA-mediated CAD knockdown. CAD siRNA (si-CAD) transfection for 48 h led to a significant reduction of CAD protein in comparison to non-transfected and control siRNA (si-Control) transfected PAH pericytes (Fig. 2h). In control pericytes, PDGF-BB increased proliferation by approximately 1.8-fold, and CAD knockdown (siCAD) only mildly reduced this proliferative response. In contrast, PAH pericytes showed a markedly enhanced

proliferation under PDGF-BB stimulation, which was significantly attenuated by siCAD (by approximately 50% (Fig. 2i)).

Together these results indicate increased DNPyS in hyperproliferating PAH pericytes in vitro (in response to PDGF-BB) and in situ. Enhanced phosphorylation and activation of CAD contributes to this pathological phenotype.

### In control and PAH pericytes CNP/GC-B/cGMP/cGKI signaling prevents PDGF-BB-induced increases of HIF-1α and GLUT-1

Unraveling pathways preventing or even reverting the above-described metabolic dysregulation of PAH pericytes may provide novel targets for the treatment of patients with this fatal disease. We reported recently that CNP, via GC-B/cGMP signaling, attenuates the hyperproliferation and migration of control and, even more, of PAH lung pericytes[16]. To investigate whether the antiproliferative effect of CNP is partly mediated by metabolic pathways, firstly we examined the impact of CNP on PDGF-BB-induced pericyte glucose uptake. As shown in Fig. 3a, b, CNP pretreatment (30 min, 100 nM) significantly attenuated PDGF-BB-induced proliferation (reiterating our published findings[16]) as well as $^{18}$F-FDG uptake of both control and PAH pericytes (PDGF-BB: 30 ng/ml, 24 h). Concomitantly, in both groups, CNP prevented the stimulatory effects of PDGF-BB on GLUT-1 mRNA and protein expression (Fig. 3c, d show results from qRT-PCR and immunoblots, respectively). The transcription factor HIF-1α is known to induce GLUT-1 expression and has been implicated in PAH pathogenesis[8,25]. Therefore, we analyzed the expression of HIF-1α in control and PAH pericytes and observed that HIF-1α protein expression is upregulated by PDGF-BB and that CNP pretreatment almost fully abolished this effect (Fig. 3d).

Which mechanisms mediate this interaction between CNP and PDGF-BB? As schematized in Fig. 3e, PDGF-BB, via PI3K/AKT signaling, activates the mammalian target of rapamycin complex 1 (mTORC1) pathway, which enhances the translation of HIF-1α mRNA even under non-hypoxic conditions[24,25]. Indeed, as shown in Fig. 3f, pretreatment of control pericytes with the AKT inhibitor, Akti-1,2 (5 μM, 20 min), or the mTORC inhibitor, Rapamycin (100 nM, 20 min), prevented the induction of HIF-1α and GLUT-1 by PDGF-BB. On the other hand, our recent studies in human lung pericytes revealed that CNP, via cGKI, inhibits the PDGF-BB-PI3K-AKT pathway via activation of Phosphatase and tensin homolog (PTEN)[16]. Thus, we hypothesized that CNP inhibits PDGF-BB-induced HIF-1α/GLUT-1 expression by cGKI-dependent AKT/mTORC1 inactivation (see left part of scheme in Fig. 3e). To study the participation of cGKI, we tested CNP in the presence of a specific inhibitor. Pretreatment of pericytes with Rp-8-Br-PET-cGMPS (10 μM, 20 min), abolished the inhibitory effect of CNP on the PDGF-BB-induced HIF-1α and GLUT-1 expression (Fig. 3g).

Together with our previous findings[16], these new observations indicate that CNP inhibits PDGF-BB-induced HIF-1α and GLUT-1 expression in control and PAH pericytes through cGMP/cGKI-dependent inactivation of the AKT/mTORC1 pathway.

**Fig. 2 | PAH pericytes demonstrate increased de novo pyrimidine synthesis and CAD protein phosphorylation. a** Metabolic pathways enriched in PAH pericytes under PDGF-BB stimulation versus controls based on quantitative metabolite set enrichment analysis using MetaboAnalyst 6.0. The corresponding fold enrichments and computed p values are depicted. **b** Fold change analysis using Metaboanalyst 6.0 shows significantly upregulated metabolites in PAH pericytes in comparison to control cells under baseline (**b**, upper panel) and PDGF-BB stimulation (**c**, lower panel). Red circles represent metabolites above the threshold. **c** Schematic representation of $^{13}C$ labeling patterns after the metabolism of $^{13}C_6$-glucose through de novo pyrimidine synthesis. **d** Normalized peak areas of $^{13}C$-labeled pyrimidine metabolites, as measured by targeted LC-MS, in control and PAH pericytes stimulated with PDGF-BB and labeled with $^{13}C_6$ glucose depicting upregulated (M + 5) and (M + 2) labeled pyrimidines and unchanged (M + 1) labeled pyrimidines (n = 3 controls and PAH, unpaired 2-tailed Student's t test). **e** Schematic illustration highlighting the investigated enzymes. Created with Biorender.com. **f** CAD phosphorylation at Threonine 456 was significantly increased in cultured PAH pericytes compared to controls with no difference in CAD protein expression. (n = 3 controls and n = 4 PAH patients; unpaired 2-tailed Student's t test). **g** Representative immunofluorescence staining of pCAD$_{Thr456}$ on control and IPAH patient lung tissues, followed by fluorescence quantification from n = 4 tissues from each group. pCAD$_{Thr456}$ (white), PDGFR-β (red: pericyte marker), ULEX (green: endothelial stain), and nuclei were stained by DAPI (blue). Arrows indicate the colocalization of PDGFR-β with pCAD$_{Thr456}$. Scale bar = 20 μm. **h** Transfection of PAH pericytes with siCAD reduced CAD protein expression (n = 4 biological replicates; 1-way ANOVA), and (**i**) this inhibited PDGF-BB induced proliferation in PAH pericytes but not in control cells (n = 4 control and PAH pericytes; 2-way ANOVA). For **d**: *p < 0.05 vs Controls – PDGF-BB. For **f** and **g**: *p < 0.05 vs Controls. For **h**: *p < 0.05 vs. untransfected control (−), #p < 0.05 vs. si-Control. For **i** *p < 0.05 vs. PBS (−), #p < 0.05 vs. si-Control-PDGF-BB, $p 0.05 vs. similar treatment. Data is presented as Mean ± SEM.

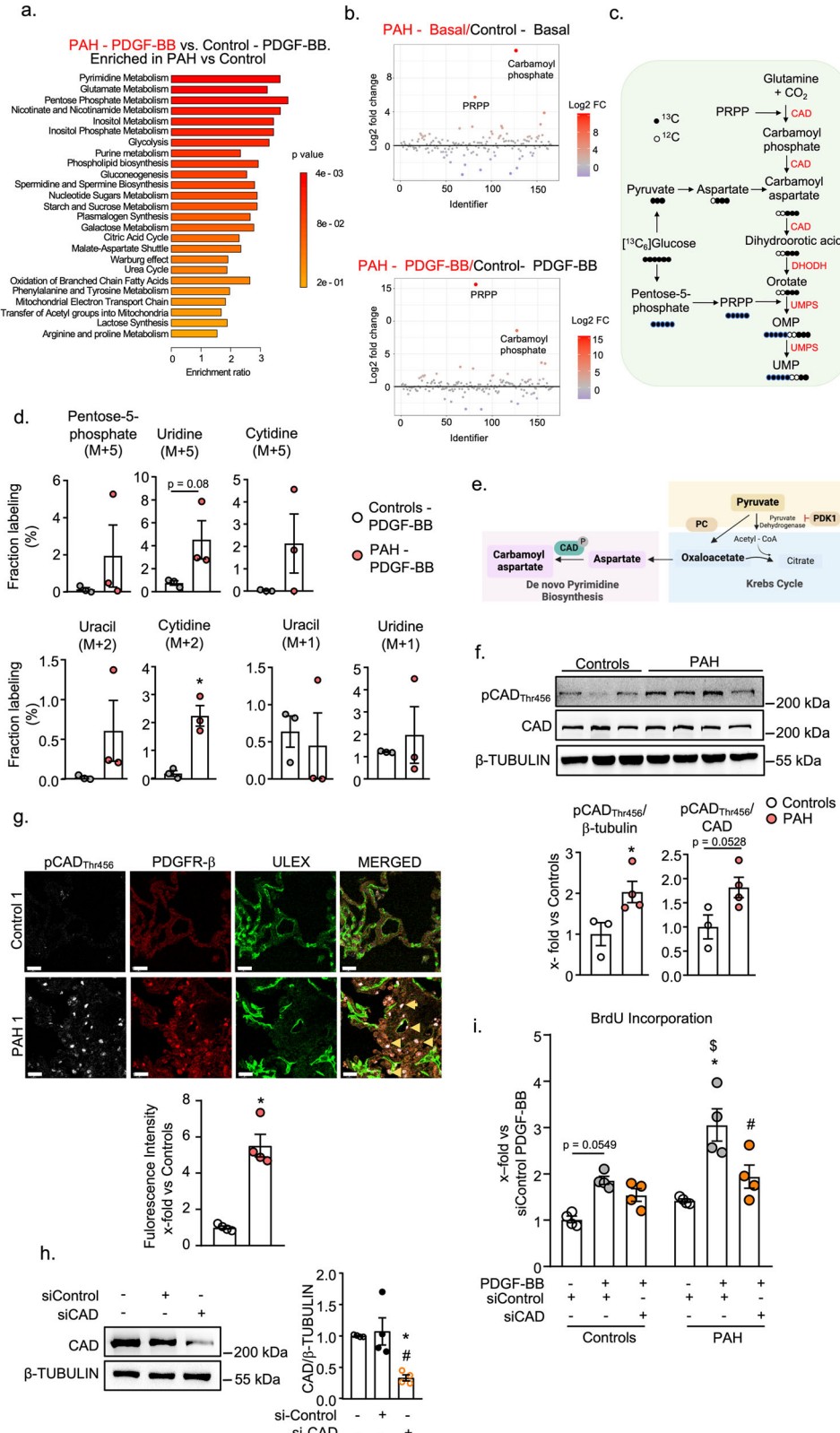

### In PAH pericytes CNP inhibits PDK1 expression and CAD phosphorylation and thereby PDGF-BB-induced pyruvate conversion to lactate and aspartate

Next, we analyzed whether CNP counter-regulates PDGF-BB-induced DNPyS in PAH pericytes. Interestingly, the PDGF-BB-induced increase in pyruvate (M + 3), the end product of glycolysis, was reduced by CNP

pretreatment in PAH but not in control pericytes (Fig. 4a, mole fractions in Supplementary Fig. 6a). This distinct metabolic response to CNP may relate to intrinsic differences in pyruvate kinase isoform (PKM1 and PKM2) expression, where a higher PKM2/PKM1 ratio favors the Warburg effect as has been observed in cancer cells as well as PAH adventitial fibroblasts[26,27]. Indeed, PAH pericytes expressed lower PKM1 levels, resulting in a high

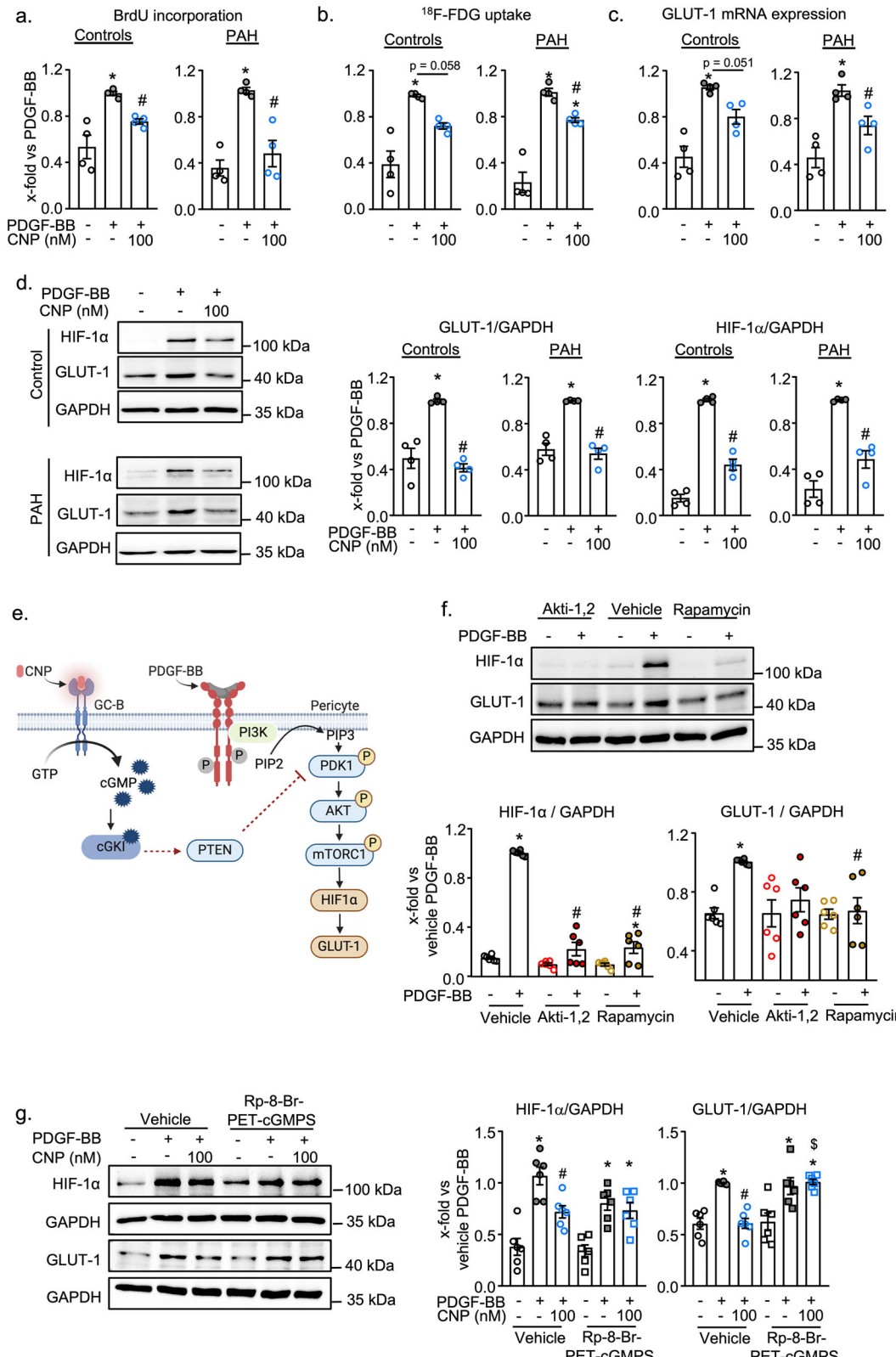

PKM2/PKM1 ratio (Supplementary Fig. 5a, b). Our future studies will explore whether CNP regulates PKM1 and PKM2 expression and activity in PAH pericytes thereby lowering the PKM2/PKM1 ratio.

Pyruvate can fuel four different pathways (Scheme in Fig. 4b). It can be converted to lactate by lactate dehydrogenase, to acetyl-CoA by pyruvate dehydrogenase (which enters the TCA cycle), or to alanine by alanine transaminase[23]. Additionally, pyruvate carboxylase converts pyruvate to oxaloacetate which is converted to aspartate, to fuel DNPyS[23] (Fig. 4b). As already mentioned, the $^{13}C_2$-labeling (M + 2) of the intermediates of the TCA cycle was very weak. Further, PDGF-BB stimulation, with or without CNP pretreatment, did not alter the oxygen consumption rate (OCR) in either control or PAH pericytes (Supplementary Fig. 6b). Therefore, instead

**Fig. 3 | CNP prevents PDGF-BB induced glucose uptake and glucose transporter, GLUT1 expression in a cGKI - HIF-1α dependent manner.** CNP (100 nM, 30 min pretreatment) prevented PDGF-BB (30 ng/ml, 24 h) induced (**a**) proliferation, (**b**) glucose uptake and (**c**) GLUT-1 mRNA expression in control (left panels) and PAH pericytes (right panels). (**a**: BrdU incorporation; **b**: $^{18}$F-FDG glucose uptake; **c**: qRT PCRs. n = 4 from controls and PAH, 1-way ANOVA). **d** Immunoblotting: CNP (100 nM, 30 min) prevented the PDGF-BB (30 ng/ml, 24 h)—induced GLUT-1 and its regulator, HIF-1α expression in control and PAH pericytes (n = 4 control and PAH, 1-way ANOVA). **a** The new experiments illustrated in this panel reproduce findings from our earlier study (Dabral S et al. Commun Biol. 2024) for comparison and validation. **e** Scheme of the postulated and investigated signaling pathway. Created with Biorender.com (https://BioRender.com/3ripqtn). **f** AKT inhibitor (Akti-1, 5 μM) and mTORC inhibitor (Rapamycin, 100 nM) prevent PDGF-BB induced HIF-1α and GLUT-1 expression in control pericytes (n = 6; 1-way ANOVA). **g** cGKI inhibitor Rp-8-Br-PET-cGMPS (10 μM), prevented the effect of CNP on PDGF-BB (30 ng/ml, 24 h) - induced HIF-1α and GLUT-1 expression (n = 6; 2-way ANOVA). For **a, b, c, d,** and **f**: *$p < 0.05$ vs. PBS (−), #$p < 0.05$ vs. si-Control-PDGF-BB. For g: *$p < 0.05$ vs PBS (−), #$p < 0.05$ vs PDGF-BB, $p < 0.05$ vs corresponding vehicle-treated group (−). Data is presented as Mean ± SEM.

of the TCA cycle we investigated the effects of CNP on enrichment of pyruvate-derived lactate, alanine, and aspartate. The levels of intracellular lactate (M + 3) was not changed by PDGF-BB (Fig. 4c). However, the ECAR, an indicator of glycolysis and extracellular lactate, was significantly increased by PDGF-BB in both control and PAH pericytes (Fig. 4d, mole fractions in Supplementary Fig. 6c). Notably, CNP prevented this response only in PAH pericytes and not in controls (Fig. 4d). As shown in Fig. 4e, f, PDGF-BB also increased the levels of alanine (M + 3) and aspartate (M + 3) in PAH pericytes. CNP mildly altered the effect of PDGF-BB on alanine labeling, but significantly reduced the effect on aspartate labeling (Fig. 4e, f, mole fractions in Supplementary Fig. 6c). Corroborating these findings, fold change (FC) analyses of the metabolomics data revealed that CNP pretreatment significantly prevented the stimulatory effects of PDGF-BB on the levels of carbamoyl aspartate, deoxy-Cytidine and especially of carbamoyl phosphate in PAH pericytes (Fig. 4g). Figure 4h remarks that PDGF-BB and CNP modulated the carbamoyl aspartate levels exclusively in PAH pericytes and not in control cells.

Next, we studied whether the inhibitory effect of CNP on DNPyS of PAH pericytes involves changes in PDK1 and CAD expression or CAD phosphorylation. PDGF-BB significantly enhanced PDK1 and CAD expression as well as CAD$_{Thr456}$ phosphorylation (Fig. 4i). As also shown, CNP fully abolished this effect. To investigate the *acute* effects of CNP and PDGF-BB on CAD$_{Thr456}$ phosphorylation, PAH pericytes were treated with CNP (100 nM) for 30 min and then with PDGF-BB for an additional 30 min. As shown in Fig. 4j, CNP completely abolished the acute stimulatory effect of PDGF-BB on CAD$_{Thr456}$ phosphorylation. In control pericytes, PDGF-BB also enhanced the expression of PDK1 and CAD phosphorylation but did not increase CAD expression (24 h and 30 min stimulation) (Supplementary Fig. 7a, b). Importantly, CNP inhibited PDGF-BB induced PDK1 expression however, had no effect on CAD phosphorylation indicating that this later pathway is responsive to CNP only in PAH pericytes.

In conclusion, in PAH pericytes CNP inhibits PDGF-BB-induced DNPyS both by limiting pyruvate availability and by modulating CAD phosphorylation.

## In PAH pericytes CNP prevents PDGF-BB-stimulated CAD phosphorylation via activation of phosphodiesterase 2 and inhibition of cAMP/EPAC/MEK signaling

The observation that CNP inhibits PDK1 expression in control and PAH pericytes but CAD phosphorylation exclusively in the latter, led us to hypothesize that the inhibitory effects of CNP on these two metabolic enzymes are mediated by distinct pathways.

PDK1 is a direct HIF-1α target[28], suggesting its regulation by CNP/cGKI-mediated AKT-mTORC1 inactivation, as already shown for GLUT-1 (see Fig. 3). This hypothesis is supported by three findings (Supplementary Fig. 8): (i) PDK1 mRNA expression was upregulated by PDGF-BB in both control and PAH pericytes and this was prevented by CNP (Supplementary Fig. 8a); (ii) specific inhibitors of AKT and mTORC prevented the stimulatory effect of PDGF-BB on PDK1 protein expression (Supplementary Fig. 8b); and (iii) inhibition of cGKI (10 μM Rp-8-Br-PET-cGMPs, 20 min pretreatment), abolished the inhibitory CNP effect on PDGF-BB induced PDK1 expression (Supplementary Fig. 8c).

Which pathway mediates the inhibitory CNP effect on CAD phosphorylation? Because the cGMP responses to CNP were similar in both control and PAH pericytes[16], it is unlikely that differences in receptor activation account for the observed divergence of downstream effects. Apart from from cGKI, CNP, via cGMP, can regulate the activity of two cAMP hydrolyzing phosphodiesterases (PDEs): PDE3 is inhibited by cGMP whereas PDE2 is stimulated by cGMP[12]. Thereby, CNP, via cGMP, can either enhance or decrease intracellular cAMP levels. Interestingly, while the expression levels of cGKI and PDE3A were unaltered in PAH pericytes[16], the expression of PDE2A was markedly upregulated (Fig. 5a). CNP (100 nM, 10 min) significantly decreased the baseline cAMP levels of PAH pericytes and this effect was prevented by the specific PDE2 inhibitor BAY 60-7550 (100 nM, 20 min pretreatment) (Fig. 5b). Moreover, BAY 60-7550 abolished the inhibitory effects of CNP on PDGF-BB-induced proliferation of PAH pericytes (Fig. 5c). Together these results indicate that in PAH pericytes CNP stimulates a PDE2-mediated negative cGMP-cAMP crosstalk, which participates in its antiproliferative effects. Does this pathway also mediate the inhibitory effect of CNP on PDGF-BB-stimulated CAD phosphorylation? Is the stimulatory effect of PDGF-BB partly mediated by cAMP signaling?

PDGF-BB enhanced cAMP levels in vascular cells and cultured PCLS, possibly through the release and autocrine action of prostaglandin (PG) E2[29,30]. As shown here, PDGF-BB (30 ng/ml, 5 min) also increased cAMP levels in cultured lung pericytes, and this effect was attenuated by CNP pretreatment (cAMP ELISA, Fig. 5d). In general agonists enhancing cAMP levels, through activating the exchange protein EPAC1, can trigger phospholipase C (PLC)—ε and protein kinase C (PKC) signaling[31], and thereby MAPK/ERK1/2 activity (scheme in Fig. 5e). ERK1/2 phosphorylates CAD at Thr$_{456}$ thereby augmenting its activity and its sensitivity to the allosteric activator, Phosphoribosyl pyrophosphate[27]. To elucidate whether this EPAC1/ERK1/2 pathway mediates CAD phosphorylation by PDGF-BB, PAH pericytes were pretreated with the EPAC inhibitor, ESI-09 (1 μM, 20 min) or the MEK inhibitor, PD98059 (10 μM, 20 min). As shown in Fig. 5f, PDGF-BB (30 min) markedly increased ERK1/2 and CAD phosphorylation and both effects were significantly attenuated by these inhibitors. Notably, the EPAC inhibitor even abolished baseline CAD phosphorylation (Fig. 5f). This might be related to attenuation of EPAC-dependent basal PKC activity, thereby reducing CAD$_{Ser1873}$ and concomitant CAD$_{Thr456}$ phosphorylation[32,33]. As there is no commercially available antibody for detecting the phosphorylation of CAD$_{Ser1873}$, this possibility could not be experimentally tested.

Does CNP/PDE2 signaling interfere with this PDGF-BB-driven molecular pathway? As illustrated in Fig. 5g, CNP pretreatment reduced the stimulatory effects of PDGF-BB not only on CAD$_{Thr456}$ but also on ERK1/2$_{Thr202/Tyr204}$ phosphorylation. The PDE2 inhibitor BAY 60-7550 abolished such inhibitory effects of CNP on PDGF-BB-induced CAD and ERK phosphorylation.

Together, these findings suggest that CNP, via cGMP, regulates CAD phosphorylation and PDK1 expression in PAH pericytes through two distinct pathways. In future studies, we will elucidate the mechanism underlying the effects of PDGF-BB and CNP on CAD protein expression in PAH pericytes.

**Fig. 4 | CNP prevents PDGF-BB induced pyruvate utilization for lactate and aspartate in PAH pericytes by regulation of PDK1 expression and CAD phosphorylation. a** Schematic overview of fate of pyruvate (M + 3). **b** CNP pretreatment reduces PDGF-BB induced increase in pyruvate (M + 3) in PAH pericytes, with no effect in control pericytes. (n = 3 controls and PAH; 1–way ANOVA). **b** PDGF-BB has no effect on intracellular lactate (M + 3) in PAH pericytes (n = 3 PAH; 1–way ANOVA). **c** CNP pretreatment reduces PDGF-BB induced aerobic glycolysis only in PAH pericytes, with no effect in control pericytes. (n = 3 control and 4 PAH; 1 – way ANOVA) (**e** and **f**) PDGF-BB stimulated increase in (**e**) alanine (M + 3) and (**f**) aspartate (M + 3) and CNP prevented this effect only on aspartate. (n = 3 PAH; 1 – way ANOVA). **g** Fold change analysis using Metaboanalyst 6.0 shows significantly downregulated metabolites in CNP pretreated PAH pericytes in comparison to PDGF-BB stimulated PAH cells. Blue circles represent metabolites above the threshold. **h** CNP reduced PDGF-BB induced carbamoyl aspartate metabolite levels in PAH pericytes but not in controls (n = 3 control and PAH; 1 – way ANOVA). **i** Pretreatment with CNP prevented the PDGF-BB (30 ng/ml, 24 h) induced CAD phosphorylation (Thr456) as well as CAD and PDK1 expression. **j** CNP prevented PDGF-BB (30 ng/ml, 30 min) induced CAD phosphorylation (for **i** and **j**: n = 4 biological replicates; 1-way ANOVA). For **b, c, d, e, f, h, I**, and **j**: *p < 0.05 vs. PBS (–), #p < 0.05 vs⁻ PDGF-BB. Data is presented as Mean ± SEM.

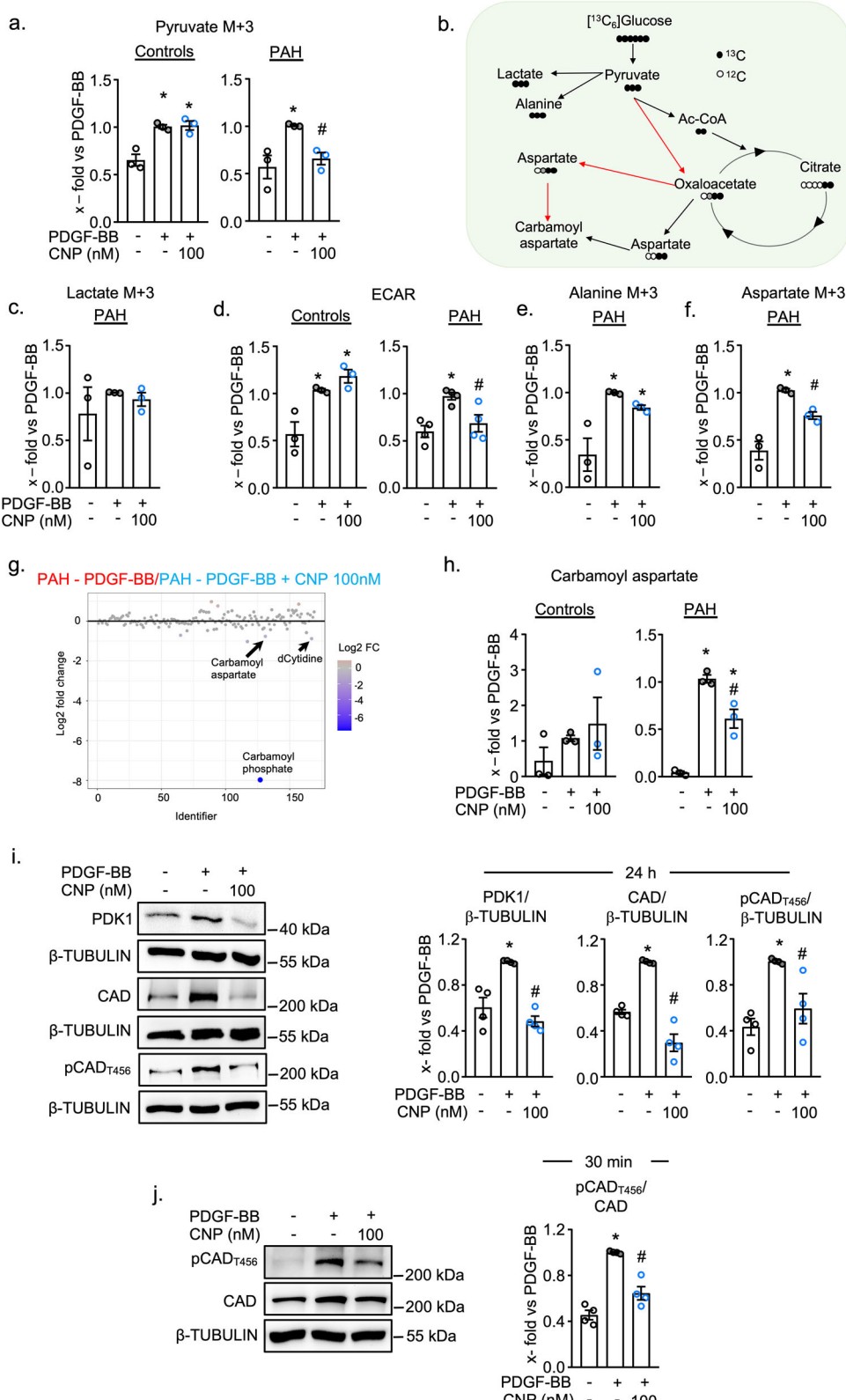

## CNP prevents PDGF-BB-induced proliferation and metabolic gene expression of murine pericytes in situ, in cultured precision cut lung slices

To study CNP effects on lung pericytes in situ, we used the ex vivo model of cultured murine PCLS. As illustrated in Fig. 6a, the slices were cultured in the presence of PDGF-BB (30 ng/ml, replenished every 2nd day) during 7 days, with/without CNP (100 nM). At the end of the experiment PCLS cryosections were immunostained with antibodies against the pericyte marker protein NG2, followed by fluorescence signal quantification with Image J; in addition, proteins were extracted for immunoblot analyses. As shown in Fig. 6b, PDGF-BB significantly increased the number of NG2+ pericytes as well as overall NG2 fluorescence intensity per field. These

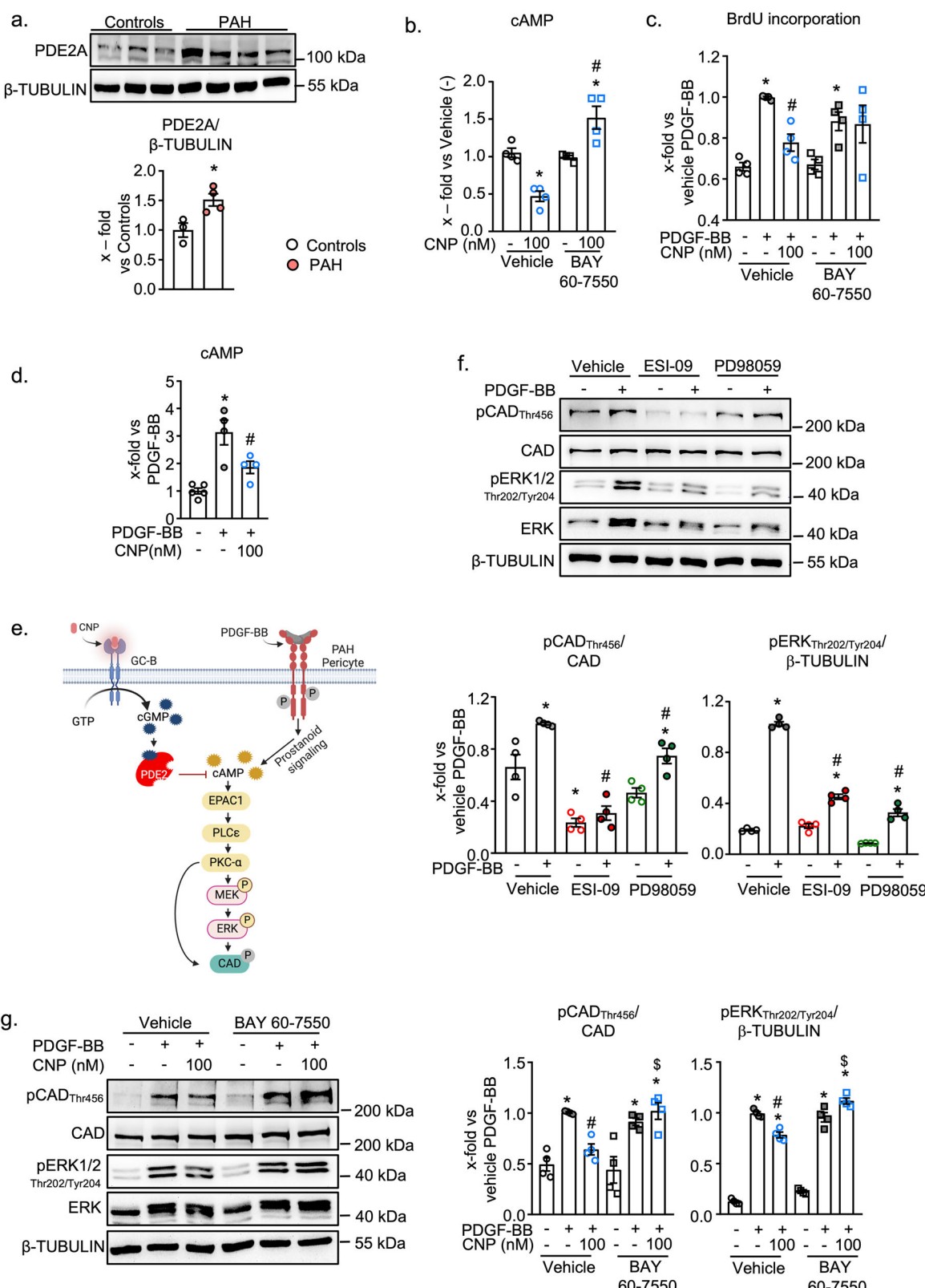

effects were markedly prevented by CNP (representative images in Fig. 6b; quantitative analyses in Fig. 6c, upper panel). To elucidate whether such changes were associated with pericyte proliferation, we performed co-immunostainings of NG2 and proliferating cell nuclear antigen (PCNA), and counted double positive (NG2+, PCNA+) cells. After 7 days of PDGF-BB treatment the number of PCNA+ pericytes per

field was markedly increased and this effect was prevented by CNP (Fig. 6c, lower panel).

Our immunoblot studies aimed to elucidate whether the metabolic pathways dissected in cultured PAH pericytes were also regulated by PDGF-BB and CNP in murine pericytes in situ. In line with the immunostainings, PCNA was strongly induced in PDGF-BB-treated lung sections and this

**Fig. 5 | CNP prevents CAD phosphorylation via PDE2 mediated reduced cAMP/EPAC/MEK signaling in PAH pericytes. a** Lung pericytes isolated from PAH patients exhibit higher PDE2A expression (n = 3 control pericytes, n = 4 PAH pericytes; unpaired 2-tailed Student's t test). **b** CNP (10 min) significantly reduced cAMP levels which was reversed by PDE2 inhibitor, BAY 60-7550 (100 nM, 20 min pretreatment) (n = 4 PAH, 1-way ANOVA). **c** BAY-60-7550 (PDE2 inhibitor, 100 nM) attenuated the inhibitory effects of CNP on PDGF-BB-induced proliferation of PAH pericytes as analysed by BrdU incorporation assay (n = 4 biological replicates; 2-way ANOVA). **d** PDGF-BB (5 min) significantly increased cellular cAMP levels which was attenuated by 10 min CNP 100 nM pretreatment (n = 4 PAH, 1-way ANOVA). **e** Scheme of the postulated and investigated signaling pathway. Created in BioRender (https://BioRender.com/3ripqtn). **f** PDGF-BB (30 ng/ml, 30 min) increased CAD and ERK1/2 phosphorylation which was prevented by EPAC inhibitor (ESI-09, 1 μM) and MEK inhibitor (PD-98059, 10 μM) (n = 4 PAH pericytes; 2-way ANOVA). **g** PDE2 inhibitor BAY60-550 (100 nM) prevented the effect of CNP on PDGF-BB (30 ng/ml, 30 min)-induced phosphorylation of CAD (Thr456) and ERK1/2 (Thr202/Tyr204) (n = 4 PAH pericytes; 2-way ANOVA). For **a**: *$p < 0.05$ vs Controls. For **b**: *$p < 0.05$ vs. PBS (−), #$p < 0.05$ vs. Vehicle⁻ CNP 100 nM. For **c, d, f**: *$p < 0.05$ vs. PBS (−), #$p < 0.05$ vs. PDGF-BB. For **g**: *$p < 0.05$ vs PBS (−), #$p < 0.05$ vs PDGF-BB, $$p < 0.05$ vs corresponding vehicle-treated group (−). Data is presented as Mean ± SEM.

induction was inhibited by CNP (Fig. 6d). Concomitantly, PDGF-BB-stimulated CAD phosphorylation at $Thr_{456}$ as well as the expression of HIF-1α, GLUT-1, CAD, and PDK1. Notably, CNP significantly prevented all these effects (Fig. 6d).

Taken together, the results from these studies of murine PCLS corroborate the findings in cultured human lung pericytes demonstrating that CNP effectively reduces PDGF-BB-induced pericyte proliferation and the associated upregulation of metabolic pathways. These findings indicate that CNP may counteract the metabolic and proliferative alterations of lung pericytes in PAH.

## Discussion
### Summary
By combining stable isotope metabolic tracing with steady state metabolomics in vitro, we demonstrate that enhanced glycolysis and augmented de novo pyrimidine synthesis (DNPyS) both contribute to the hyperproliferative phenotype of lung pericytes from PAH patients. GLUT-1-mediated glucose uptake and glycolysis as well as CAD-driven DNPyS are increased in PAH pericytes to meet their heightened metabolic demands, maintaining excessive growth. Notably, CNP, via GC-B/cGMP signaling, attenuates these changes by inhibiting HIF-1α, AKT, and ERK1/2 activities, thereby suppressing PAH pericyte proliferation. Notably, these protective effects of CNP were also observed in activated pericytes in situ, in murine PCLS. Our findings stimulate investigating the therapeutic potential of recently developed stabilized, "long acting" CNP analogs[47] in experimental and clinical PAH.

### Increased GLUT-1 expression and enhanced glycolysis contribute to hyperproliferation of PAH pericytes
In PAH, glucose metabolism of vascular endothelial and smooth muscle cells shifts from mitochondrial respiration towards higher glycolysis, the "aerobic Warburg effect"[3,8,34]. This shift is similar to that observed in cancer cells and contributes to the pathogenesis of PAH by promoting vascular cell growth and thereby vascular thickening[35]. The here presented studies reveal for the first time that lung pericytes derived from PAH patients also exhibit increased glucose uptake and glycolysis. Consistent with this, metabolomic studies carried out in PAH pulmonary artery (PA) endothelial cells (PAECs), PA smooth muscle cells (PASMCs) and adventitial fibroblasts have shown increased glucose uptake and augmented levels of glycolytic intermediates in diseased cells compared to controls, supporting glycolysis as a shared metabolic feature across multiple vascular cell types in PAH patients[10,36,37]. This increase is linked to augmented expression of the glucose transporter, GLUT-1. In our study, inhibition of GLUT-1 activity attenuated PDGF-BB-induced pericyte glucose uptake and proliferation. Augmented GLUT-1 expression was also observed in adventitial fibroblasts and endothelial cells of PAH patients[34,38]. Together, these finding highlight GLUT-1 upregulation as a common mechanism in diseased pulmonary vascular cells, which not only enhances glucose uptake to meet the increased energy demands but also further fosters cell hyperproliferation.

### Increased CAD expression and activity trigger de novo pyrimidine synthesis in PAH pericytes
A striking additional finding of this study is the upregulation of DNPyS in PAH pericytes. Targeted metabolomics identified pyrimidine metabolism

as the most enriched metabolic pathway in PDGF-BB-treated PAH pericytes, in line with a significant upregulation of DNPyS intermediates in FC analyses. Isotope labeling studies confirmed a markedly increased pool of pyrimidine nucleotides derived from PRPP and anaplerotic conversion of pyruvate in PAH pericytes. Consistent with these findings, the activity of CAD, the enzyme catalyzing the rate limiting step in DNPyS, and the expression of PDK1, which inhibits the entry of pyruvate to TCA cycle, were increased. Such alterations were observed both in cultured human PAH pericytes as well as in PDGF-BB-treated murine lung pericytes in situ (in PCLS). The selective antiproliferative effect of siRNA-driven CAD knockdown in PAH pericytes highlights a preferential targeting of a disease specific metabolic alteration. RNA-sequencing analysis of pulmonary artery adventitial fibroblasts from calves with PAH had already shown upregulated pyrimidine metabolism[10]. However, to the best of our knowledge, this is the first study to demonstrate the exacerbated DNPyS in human PAH pericytes. Both purine and pyrimidine nucleotides are indispensable for DNA/RNA synthesis and cell division. Nucleotide synthesis is essential for rapidly dividing cells, as salvage pathways are insufficient to meet proliferative demands. Thus, upregulation of de novo purine and pyrimidine synthesis is a well-established feature of cancer and other hyperproliferative diseases[39,40]. In the context of vascular remodeling in PAH, Ma et al. reported that de novo purine synthesis, driven by the enzyme ATIC, contributes to the hyperproliferative phenotype of PASMCs[5]. In the same direction, recent work has shown that ATP-citrate lyase (ACLY), which cleaves cytosolic citrate to acetyl-CoA and oxaloacetate, is upregulated in pulmonary arteries from PAH patients[41]. ACLY-derived acetyl-CoA supports lipid biosynthesis and epigenetic modifications while oxaloacetate may be converted to aspartate, feeding into DNPyS via CAD[32]. Thereby, ACLY is a key metabolic node linking lipid and nucleotide synthesis.

Recent studies have further highlighted the complexity of PAH-associated metabolic rewiring, implicating glutamine/serine metabolism[42], de novo lipid and cholesterol synthesis, and acetyl-CoA–mediated epigenetic regulation[41] in disease progression. Although our study did not directly assess glutamine or serine metabolism, these pathways are metabolically interconnected with glycolysis and nucleotide synthesis. Glutamine provides nitrogen and carbon for nucleotide biosynthesis[43], while serine contributes one-carbon units for purine/pyrimidine synthesis and NADPH for redox homeostasis[44]. Both depend on glycolytic intermediates such as 3-phosphoglycerate and glucose-derived carbon backbones, which are upregulated in PAH pericytes. Further investigation is warranted to delineate the interplay between these interconnected pathways.

### CNP prevents the metabolic remodeling of PAH pericytes through activation of cGKI and PDE2
The pathologic phenotype of vascular cells in PAH can be partly reversed by targeting metabolic changes. For instance, the PDK inhibitor, dichloroacetate (DCA), increased glucose oxidation and attenuated excessive proliferation and apoptosis resistance in PASMCs in vitro[45]. Moreover, it improved hemodynamic and functional capacity of patients with idiopathic PAH in a 4-month Phase 1 clinical trial[46]. However, patients with inactivating mutations in the genes encoding Sirtuin 3 and Uncoupling protein 2 did not respond to DCA[46]. In our study, treatment of PAH pericytes with DCA reduced PDGF-BB-induced lactate levels and cell proliferation, but

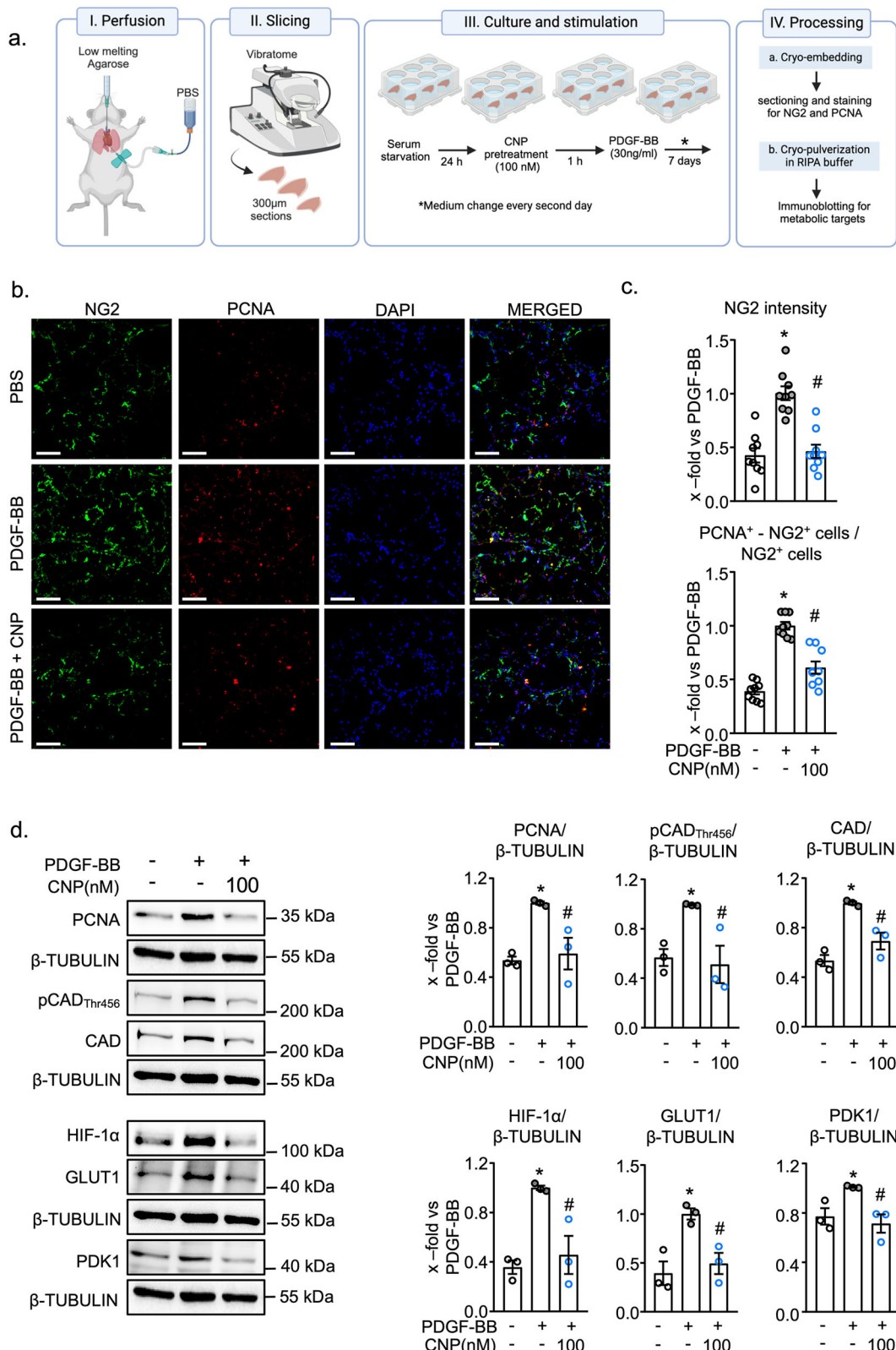

**Fig. 6 | Exogenous CNP attenuates PDGF-BB induced pericyte proliferation and metabolic gene expression in ex vivo cultured murine precision cut lung slices (PCLS). a** Schematic overview of experimental design. Created with Biorender.com (https://BioRender.com/bjgvpyp). **b** Representative images of NG2 (green) and PCNA (red) in murine lung sections (Scale bar 100 μm). Images were acquired with Leica SP8 confocal microscope and processed with Image J. **c** CNP (100 nM) prevented PDGF-BB (30 ng/ml) mediated increase in pericyte proliferation as analyzed by co-staining of Neural/glial antigen 2, NG2 (pericyte marker) with proliferating cell nuclear antigen, PCNA (proliferation marker). (n = 9 from 3 PCLS from 3 mice. Each value is a mean of 3 images; 1-way ANOVA). **d** PDGF-BB stimulated CAD phosphorylation and expression of PCNA, CAD, HIF-1α, GLUT-1 and PDK1 in PCLS which was prevented by CNP (100 nM) (n = 3 PCLS from 3 mice; 1-way ANOVA). For **c**, **d**: *$p < 0.05$ vs PBS, #$p < 0.05$ vs PDGF-BB. Data is presented as Mean ± SEM.

did not affect DNPyS. Hence, instead of therapies that target isolated metabolic steps, a pharmacological approach addressing multiple dysregulated pathways may offer greater therapeutic benefit.

Our novel findings suggest that CNP may prevent the altered metabolism of PAH pericytes at different steps and through several cGMP-regulated pathways. We previously showed that CNP/GC-B signaling counteracts hyperproliferation, excessive migration, and differentiation of cultured PAH pericytes[16]. This study adds an important mechanistic information showing that CNP reduces the stimulatory effects of growth factors such as PDGF-BB on GLUT-1 expression and glucose uptake in both control and PAH pericytes, thereby attenuating glycolysis. Simultaneously, CNP attenuated PDK1 and CAD expression as well as the activating CAD-Thr$_{456}$ phosphorylation in PAH pericytes, thereby reducing DNPyS. This is further evidenced by decreased levels of aspartate (M + 3), derived from pyruvate anaplerosis, and of carbamoyl aspartate, carbamoyl phosphate, and dCytidine in CNP treated PAH pericytes. Hence, CNP targets the metabolic dysregulation of PAH pericytes at multiple steps. This broader targeting likely explains the stronger anti-proliferative effect of CNP in PAH pericytes ( ~ 75%), compared to DCA ( ~ 18%), which impacts only glycolysis but not DNPyS.

The present work also investigated the mechanisms underlying the distinct metabolic effects of CNP in PAH versus control pericytes. CNP/GC-B, via cGMP, can activate cGKI and/or the cGMP/cAMP-degrading PDE2. While cGKI expression and activity were similar in control and PAH pericytes[16], PDE2A expression was significantly increased in the latter. Thus, we hypothesized that the metabolic effects of CNP observed in both control and PAH pericytes were mediated by cGKI, while CNP effects which occurred specifically in PAH pericytes were mediated by PDE2.

Indeed, CNP, via cGKI, attenuated PDGF-BB induced HIF-1α activation in an AKT-mTORC-dependent manner in both control and PAH pericytes. Through inhibition of HIF-1α, CNP diminished the expression of GLUT-1 and PDK1, which are well established targets of this transcription factor (Scheme in Fig. 3e). In fact, among the many dysregulated cellular pathways in PAH, increased HIF signaling is one of the major contributors to vascular cell proliferation and disease progression[4]. Under hypoxic conditions or in response to growth factors, HIF-1α stabilization promotes the transcription of genes involved in cellular proliferation, glycolysis, and vascular remodeling. Liu et al. recently showed that another HIF isoform, HIF-2α, plays a crucial role in the pericyte-to-smooth muscle-like cell transition in pulmonary hypertension, underscoring the importance of pericyte-specific HIF signaling in this disease[47]. Here, we demonstrate that CNP moderates pericyte HIF-1α expression and thereby glycolysis. In future studies, we will explore whether CNP exhibits similar effects on HIF-1α expression in other vascular cell types and/or also modulates HIF-2α expression in pericytes.

In addition, our studies indicate that in PAH pericytes only, CNP/cGMP signaling activates PDE2 and thereby lowers cAMP levels. This subsequently inhibits PDGF-BB induced ERK1/2 and CAD activation by reducing cAMP-dependent EPAC activation (scheme in Fig. 5e) (current study and[31]). Accordingly, PDE2 inhibition abolished and even reverted the inhibitory effects of CNP on pericyte cAMP levels and PDGF-BB-induced proliferation. In apparent contradiction with these observations, enhancing pulmonary vascular cAMP signaling by treatment with prostacyclin/prostaglandin I2 (PGI$_2$) analogs (Iloprost) is protective in patients with PAH[48]. PGI$_2$ displays strong vasodilatory and anti-proliferative effects by binding to the IP receptor on PASMCs. However, in addition to this receptor, PGI$_2$ analogs may also bind non-specifically to EP receptors, which can lead to side effects and reduce their effectiveness[49]. EP 1–4 are receptors for prostaglandin E2 (PGE$_2$)[49], which is released in response to PDGF-BB stimulation leading to cAMP increase[29]. Interestingly, in opposition to PGI$_2$, PGE$_2$, via enhancing cAMP levels, stimulates the proliferation of PASMCs[50]. Moreover, PGE$_2$, via EP$_4$/cAMP signaling promotes pulmonary endothelial cell activation and migration[51]. In general, how PGE$_2$ signaling via EP receptors affects pericyte functions is largely unknown. Perrot et al. showed that PGE$_2$ breaks down pericyte–endothelial cell interactions via EP1- and

EP4-dependent signaling mechanisms[52]. These published and our own observations raise the possibility that PDGF-BB stimulates pericytes´ PGE$_2$ release, which then augments their cAMP levels in auto/paracrine way. Our future research will attempt to dissect the role of prostaglandins and their receptors in the effects of PDGF-BB on pericytes. It will be interesting to elucidate whether PGI$_2$ and PGE$_2$ modulate different intracellular cAMP pools and whether CNP/PDE2A activity regulates specific subcellular cAMP compartments. We propose that in pericytes from PAH patients, PDE2 selectively reduces a cAMP pool which is linked to the PDGF-BB -induced PKC and ERK1/2 activation, thereby inhibiting CAD phosphorylation.

Notably, CNP mRNA levels were significantly reduced in the lungs of PAH patients and PH animals[16], suggesting a potential pathophysiological relevance of this pathway. Restoring this CNP deficiency with exogenous peptide may exert protective effects, as demonstrated in preclinical PH studies[53,54] and mechanistically reinforced by our present investigations. Vosoritide (VOXZOGOR), a stabilized analog of CNP is already clinically approved for the treatment of children with achondroplasia[55]. However, its short half-life (30 min) requires daily subcutaneous injections which lead to high peak plasma concentrations of ≈5 nM CNP. This carries the risk of arterial hypotension. Such limitations were solved by the development of the longer-acting CNP analog [Gln6, 14]-CNP-38. In animals, this analog had a half-life of nearly one month after subcutaneous injections[55]. Pharmacokinetic modeling in humans indicated that [Gln6,14]-CNP-38 could allow effective dosing regimens of weekly, bi-weekly, or even monthly injections[55], offering a therapeutic approach which would be feasible in PAH patients. Based on our here presented novel observations, long acting CNP analogs might impede or even revert the metabolic and functional abnormalities of pericytes which contribute to microvascular remodeling in this fatal disease.

Our study is limited by the fact that the metabolomics and isotope enrichment analyses were mostly conduced in cultured human pericytes, warranting validation in relevant in vivo disease models. Additionally, the single time-point isotope enrichment data reflect pathway engagement but do not capture dynamic flux, which requires follow-up with full metabolic flux analysis.

## Methods
### Clinical data of the PAH patients
The clinical information of the patients with PAH, from whom the lung tissues were obtained for pericyte isolations is available in a previous publication of Prof. Perez[56].

### Isolation of lung pericytes
All ethical regulations relevant to human research participants were followed. Written informed consent was obtained from each individual patient or the patient's next of kin.

Human lung pericytes from control (n = 3) and PAH lungs (n = 4) were provided by Prof. Perez. The study protocol for tissue donation was approved by the ethics committee (Panel on Medical Human Subjects) of the Stanford University (Stanford, US) in accordance with national law and with Good Clinical Practice/International Conference on Harmonization guidelines. The study protocol is called 'Donating Unneeded Lung Tissue, Removed During Surgery, and Blood, for Medical Research and the IRB # is 54182.

However, due to the relatively slow proliferation of control pericytes and limited cell numbers, we subsequently established independent pericyte isolations in our laboratory in Würzburg, Germany. Lung pericytes from human control donors with lung cancer were isolated from tumor-free parts of anatomical resection specimens from the Department of Thoracic Surgery, University Hospital Wurzburg. The study protocol for tissue donation was approved by the ethics committee/Ethikkommission der Universität Würzburg in accordance with national law and with Good Clinical Practice/ International Conference on Harmonization guidelines (Ref no. 20220831 02).

The details of the isolation protocol and characterization is provided in the previous publication[16,56]. For metabolomic profiling and target expression analyses (GLUT-1, phospho-CAD, CAD, PDK1, and PC), we used pericytes exclusively from Stanford to maintain group consistency:

Metabolomics: n = 3 control, n = 3 PAH; Expression studies: n = 3 control, n = 4 PAH.

For mechanistic studies, we used all 4 PAH pericytes, and included the control pericytes isolated in our laboratory in Würzburg to increase the number of independent biological replicates. As the control pericytes were derived from two different centers, which may introduce variability; we acknowledge this as a limitation when interpreting group comparisons.

### Primary cell culture
Lung pericytes were maintained in Pericyte medium (sc-1201, Provitro) supplemented with growth factors and 2% fetal bovine serum. All experiments were carried out between passages 7–11 from minimum three subjects. Each experiment was repeated from minimum three different human subjects to attest for biological heterogeneity. For functional experiments (BrdU incorporation Assay, [18]F-FDG Glucose uptake assay, seahorse assay), minimum three technical replicas were used within one experiment.

### Treatment of cultured pericytes
Before experiments, cultured cells were incubated for 24 h in basal media with 0.1% FCS (serum-reduced medium). Pericytes were pre-treated with CNP (4095840: Bachem) for 30 min, followed by stimulation with 30 ng/ml PDGF-BB (100-14B: Peprotech). CNP concentration of 100 nM was chosen based on previous publications[57–59]. For BrdU proliferation assay, glucose uptake assay and seahorse analysis, PDGF-BB stimulation was carried out for 24 h. For metabolomics, after 18 h incubation, the medium was replaced by 5 mM $^{13}C_6$ (389374, Sigma Aldrich) or regular glucose containing medium with CNP (100 nM) and PDGF-BB (30 ng/ml) for 6 h. For phosphorylation studies by immunoblotting, the pericytes were lysed 30 min after PDGF-BB stimulation. Pre-treatment with different inhibitors was carried out for 20 min before CNP stimulation wherever applicable. For cAMP ELISA assay, pericytes were treated with CNP for 10 min followed by PDGF-BB for another 5 min. Pre-treatment with PDE2 inhibitor was carried out 20 min. The list of inhibitors is provided in Supplementary Table 1.

### Metabolomics analysis (HPLC)
24 h after stimulations, cells were lysed with 1 ml of 80% MeOH/$H_2O$ (80/20, v/v), collected into tubes and snap frozen in the liquid nitrogen. Additionally, 20 μl of cell lysates were collected into separate tube for the protein determination.

25 μL of 0.2 μM lamivudine (internal reference) was added to the cell lysates, mixed, and sonicated. The lysates were centrifuged, and the supernatant was loaded onto an RP18 SPE-column, which was pre-activated with 1 ml acetonitrile and equilibrated with 1 ml MeOH/$H_2O$ (80/20, v/v). The resulting eluate was evaporated at room temperature using a vacuum concentrator and the dried samples were redissolved in 75 μL of 5 mM NH4OAc in acetonitrile/$H_2O$ (50/50, v/v).

The equipment used for LC/MS analysis was a Thermo Fisher Scientific Dionex Ultimate 3000 UHPLC system hyphenated with a Q Exactive mass spectrometer equipped with a heated electrospray ionization (HESI) probe (Thermo Fisher Scientific). LC parameters were as follows: mobile phase A consisted of 5 mM $NH_4OAc$ in acetonitrile/$H_2O$ (40/60, v/v) and mobile phase B consisted of 5 mM NH4OAc in acetonitrile/$H_2O$ (95/5, v/v). Chromatographic separation was achieved by applying 3 μl sample to the BEH Amide column (2.5 μm particles, 100 × 2.1 mm) using a linear gradient of mobile phase A and mobile phase B. The LC gradient program was 100% solvent B for 2 min, followed by a linear decrease to 10% solvent B within 23 min, then maintaining 10% B for 16 min, then returning to 100% B in 2 and 7 min 100% solvent B for column equilibration before each injection. The column temperature was set to 30 °C and the flow rate was maintained at 200 μl/min. The eluent was directed to the ESI source of the Q Exactive mass spectrometer from 2.6 – 38 min after sample injection. MS

scan parameters were as follows: scan type: full MS, scan range: 69.0–1000 m/z, resolution: 70,000, polarity: positive and negative, alternating, AGC-target: $3 \times 10^6$, maximum injection time: 200 ms HESI. Source parameters were as follows: sheath gas: 20, auxiliary gas: 1, sweep gas: 0, spray voltage: 3 kV in pos. mode and 3.6 kV in neg. mode, capillary temperature: 320 °C, S-lens RF level: 50.0, Aux gas heater temperature: 120 °C. For data evaluation: peaks corresponding to the calculated monoisotropic metabolite masses (MIM ± H + ± 3 mMU) were integrated using TraceFinder V3.3 software (Thermo Fisher Scientific). The peak area of the quantitative data set was normalized to the protein concentration. Protein concentration of each sample was determined by BCA assay after resuspending the cell pellets in RIPA lysis buffer with 10% protease and phosphatase inhibitors. For the labeled data with 13C-glucose, mass isotopomer distributions (MIDs) were calculated from raw mass spectrometry data by summing the intensities of the relevant isotopologue peaks (M + 0 to M + n) for each metabolite.

Each isotopomer fraction was then normalized by dividing the intensity of each M + n peak by the total intensity of all isotopologues. Natural isotope abundance was corrected by FluxFix.

In Fig. 4b, c, e, and f, these normalized isotopomer distribution are represented as fold change vs PDGF-BB.

Principal component analysis (PCA), Quantitative metabolite set enrichment analysis (MSEA) and fold change (FC) analysis were performed by MetaboAnalyst 6.0. The quantitative data was normalized to a pooled sample from control group, log transformed (base 10), mean centered and then divided by the square root of the standard deviation of each variable (pareto scaling) for the analysis.

### $^{18}$F-FDG Glucose uptake assay
2′-deoxy-2′-[$^{18}$F]fluoro-D-glucose ($^{18}$F-FDG) uptake was utilized to study the glucose uptake. Cells were seeded in 24-well-plate at a density of 30,000 cells/well, serum starved for 24-h and -pretreated with BAY-876 (20 min) or CNP (30 min), followed byPDGF-BB for 24 h. Last 1 h of stimulation was performed in glucose-free medium. 50 μl of $^{18}$F-FDG (500 KBq/ml) was added to the medium for 5 min at 37 °C. Then, the medium was removed immediately and washed with ice cold PBS twice. The cells were lysed in 0.1 M NaOH at 37 °C for 1 h and collected into a 1.5 ml tube for their radioactivity measurement by the gamma counter Wizard2—2480.

The values were normalized to the protein concentration which were measured by Pierce 660 nm Protein Assay (Thermo Fisher) according to manufacturer's protocol.

### Extracellular acidification rate measurements
Extra Cellular Acidification Rate (ECAR) were measured using Seahorse XF96 analyzer (Seahorse Bioscience, Agilent, US) at 37 °C. 40.000 pericytes were seeded per well in Seahorse XF96 plates, serum starved for 24 h.

Glucose-free XF Assay medium was added to the cells and 10 mM Glucose (Port A), 2.5 μM oligomycin A (Port B) and 100 mM 2-deoxy-D-glucose (Port C) (all reagents from Sigma-Aldrich). The XF96 protocol consisted of four times mix (2 min) and measurement (2 min) cycles, allowing for determination of OCR/ECAR at basal and also in between injections. ECAR after glucose addition were calculated by assessing metabolic response of the cells in accordance with the manufacturer's suggestions. The measurements were normalized to the number of cells counted by PrestoBlue (Thermo Fisher).

### Subcellular fractionation and immunoblotting
Cultured pericytes were lysed with a RIPA lysis buffer (Thermo-Scientific) containing protease and phosphatase inhibitors for whole cell lysis. For membrane fractionation, subcellular fractionation kit (Nanotools) was used according to manufacturer's instructions. SDS-PAGE and immunoblotting were performed as described previously[12]. The primary antibodies used are provided in Supplementary Table 2. GAPDH and β - Tubulin were used as loading controls for total cell lysates, and $Na^+$- $K^+$ ATPase was used for

membrane fractions. Protein bands were visualized with enhanced chemi-luminescence and quantified by densitometry.

## cAMP measurements by ELISA
Lung pericytes were seeded in 24 well plates, followed by 3 h serum starvation and stimulations as mentioned above. After stimulations, cells were lysed in ice-cold ELISA buffer, centrifuged and supernatants were used for cAMP measurements according to manufacturer's protocol (cAMP SELECT ELISA assay, cayman, 501040).

## Transfection with siRNA
Human pericytes were transfected with different siRNAs using Lipofectamine 3000 Transfection Reagent (Invitrogen) in OptiMEM serum free medium. As a control, commercially available non-targeting siRNA (si-Control) was used. siRNA sequences are provided in Supplementary Table 3. 24 h after transfection, cells were stimulated for western blotting or BrdU incorporation assay.

## Measurement of proliferation
BrdU incorporation assay (Roche Diagnostics) was used to assess the effect of different treatments on proliferation according to manufacturer's instructions. Briefly, 22 h after stimulation, cells were incubated with BrdU labeling solution for 2 h. After incubation, cells were fixed for 30 min, incubated with anti-BrdU peroxidase antibody for 90 min and finally washed with saline and incubated with substrate solution until color development. Absorbance was measured at 370 nm with reference at 492 nm in a plate reader (TECAN, Germany). Proliferation of cells was plotted as a fold change of absorbance compared to PDGF-BB-stimulated cells absorbance.

## RNA isolation and mRNA expression analysis
Total RNA was isolated using TRIzol reagent (Invitrogen), according to the manufacturer's protocol. cDNA was synthesized from the total mRNA using a high-capacity cDNA reverse transcription kit (Roche) after DNase digestion (Thermo Fisher). Real-time RT-PCR was performed using a Light Cycler Instrument (Roche). The quantitative data were calculated from the kinetic curve of the PCR by interpolation with a standard curve generated using known amounts of the target DNA.

The sequences of the primers are given in Supplementary Table 4.

## Murine precision cut lung slices
We have complied with all relevant ethical regulations for animal use. The animal studies were approved by the Animal Care and Use Committee of University of Würzburg (Regierung von Unterfranken, approval number 55.2 2532-2-1188-18) and conformed to the Animal Research: Reporting of In Vivo Experiments (ARRIVE) guidelines. Precision cut lung slices (PCLS) were prepared from 5-months old male mice (n = 3) with a mixed genetic background (C57Bl6/J; 129SV). Mice were deeply anesthetized with urethane (1.6 g/kg, i.p.); the depth of the anesthesia was checked by ensuring that noxious pinch stimulation of the hind paw, the forepaw, and the ear did not evoke any motor reflexes. Mice were then euthanized by exsanguination and their trachea was immediately cannulated. Precision cut lung slices (PCLS) were prepared from 5-months old male mice (n = 3) with a mixed genetic background (C57Bl6/J; 129SV). Mice were deeply anesthetized with urethane (1.6 g/kg, i.p.); the depth of the anesthesia was checked by ensuring that noxious pinch stimulation of the hind paw, the forepaw, and the ear did not evoke any motor reflexes. Mice were then euthanized by exsanguination and their trachea was immediately cannulated. The lungs were flushed with cold HBSS through the right ventricle, followed by lung inflation by tracheal injection of 1.5 ml of 2% low melting agarose. The lungs were removed and cooled on ice, allowing the agarose to harden. The lobes were then separated and sliced into 300 μm sections using a vibratome (VT1200, Leica). The sections were cultured in 1 ml of growth medium (DMEM-F12 + 10% FCS + 0.45% Gentamicin (10 mg/ml) + 0.6%

Amphotericin B (250 μg/ml)), with medium changes every hour initially, followed by overnight incubation. Afterwards, the PCLS was starved for 24 h, then treated with 100 nM CNP for 1 h and subsequently with PDGF-BB (30 ng/ml) over 7 days. Slices were fixed in OCT™ for further analysis or lysed with RIPA for protein isolation.

## Immunofluorescence staining
Slides with cryo-sections were thawed at RT for 20 min, washed with PBS and blocked with blocking buffer (PBS containing 5% BSA, 5% goat serum and 0.5% Triton X-100).

Double immunofluorescence was performed with primary antibodies against NG2 and PCNA (Supplementary Table 2). After overnight incubation, slides were washed and incubated with the respective secondary antibodies, Alexa 555-conjugated donkey anti-rabbit and Alexa 647-conjugated goat anti-mouse IgG (1:500, Thermo fisher) for 1 h. After washing, all sections were mounted with DAPI containing fluorescent mounting medium (Biozol).

## Statistics and reproducibility
Statistical analysis was performed with GraphPad Prism Software. All data sets are presented as means ± SEM. The individual statistical tests and sample sizes for each set of data are provided in the legends of figures and tables.

All technical replicates were averaged to generate a single value per biological sample. Data were tested for normality (Shapiro-Wilk test). For normally distributed data, Student's unpaired t test for comparing two groups and ordinary one-way ANOVA for multiple comparisons with Tukey's post hoc test were employed. For grouped data, two-way ANOVA was used. For data that were not normally distributed, the non-parametric Mann–Whitney U test was used for 2-group comparison, and the Kruskal-Wallis analysis was performed for multiple groups. Difference with $p < 0.05$ between the groups was considered significant.

## Data availability
Numerical source data for all figures can be found in Supplementary Data 1. All other data that support the findings of this study are available from the corresponding author (SD) upon reasonable request.

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

## Acknowledgements
This work was supported by the Deutsche Forschungsgemeinschaft (DA 2462/1-1 and DA 2462/3-1 to S.D.; and KU 1037/12-1, KU 1037/13-1, KU 1037/16-1, and CRC1525 (project number 453989101) to M.K.) and by Else Kroener Fresenius Stiftung (2020_EKEA.131 to S.D.).

## Author contributions
M.N. designed and performed experiments, isolated lung pericytes from control lungs, analyzed data, and generated figures. A.M. performed immunohistochemistry of human lung sections. W.S. carried out the metabolomics and analyzed the data. L.K., J.D., S.A., C.M., P.-A.L., and T.H. performed experiments and/or analyzed the results. I.A. provided lung biopsies for isolation of lung pericytes. V.A.P. provided lung pericytes isolated from control and PAH lungs. M.K. reviewed and revised the manuscript and took care of funding. S.D. developed study concept, designed, and conducted experiments, analyzed the data, wrote the first draft of the manuscript, revised subsequent drafts, and managed funding. All authors have approved the final version of the manuscript.

## Funding

## Competing interests
The authors declare no competing interests.
