## [Transparent Peer Review file · Communications Biology]

C-type natriuretic peptide attenuates enhanced glycolysis and de novo pyrimidine synthesis in pericytes of patients with pulmonary arterial hypertension

Corresponding Author: Dr Swati Dabral

This manuscript has been previously submitted at another journal. This document only contains information relating to versions considered at Communications Biology.

Version 0:

Reviewer comments:

Reviewer #1

(Remarks to the Author)

The study by Noh and colleagues present a manuscript that extends work by this group in which they position CNP as a potential inhibitor of proliferative remodeling of pulmonary vascular lesions in PAH. The focus of this particular study is on metabolic reprogramming with the experiments contained within, suggesting that the antiproliferative effects of CNP are in part mediated by effects on glucose uptake and utilization for nucleotide biosynthesis. There are aspects of this manuscript and study that are interesting and/or deploy a compelling array of orthogonal methods, including the metabolic flux analyses, utilization of human primary cells from PAH and control participants, and precision lung slice culture. This enthusiasm is tempered by issues outlined below.

1) There seems to be a systematic problem with the way the statistical analyses were performed that affects many of the figures/panels. It appears that the authors have merged technical and biological replicates, which would artificially inflate statistical power. Since all of these replicates are not 'independent' the two-way ANOVA in this approach would be incorrect. This approach is likely particularly problematic given the modest nature of many of the effect sizes presented.

2) Key findings (particularly around proliferation) overlap and are almost indistinguishable from their prior manuscript on the antiproliferative effects of CNP. This is not to suggest that this is republishing of prior results, but several of the key findings essentially appear to be a biological replicate experiments of previously published work. This diminishes enthusiasm for the impact of the central finding of CNP antiproliferative effect.

3) It is not clear that the 100nm concentration of CNP used in these studies is 'physiological' or 'pathophysiological'. Given the modest nature of some of the effect sizes seen, this raises the possibility that the findings are largely in vitro artifact.

4) More clarity around details of the control and experimental cells would be critical to fully assess the data. There is a Stanford study that is referenced (n=3 control, n=5 PAH) and a German study (controls), which seems to suggest that some of the control cells may have been obtained from an entirely different study? If this is true, there is concern that cells obtained by different investigators and then subjected to different culture programs with variable passage number, may have resulted in metabolic/proliferative changes that are not indicative of the in vivo context. There is also variability in the number of PAH and control across figures that doesn't align with the methods. For example, in some instances n=4 controls are reported and n=4 PAH (not the 3 and 5 expected) and therefore what is the rationale for the variability in N and where did the different samples come from across experiments.

5) It is not entirely surprising that PDGFR stimulation would cause a range of proliferative effects and that this would require uptake of anabolic substrate and similarly that any inhibition of proliferation would at least partially reverse the effects. The

CNP effects are interesting, but greater mechanistic clarity would be achieved if there were additional manipulations of what are proposed as the key molecular drivers of the cellular phenotypes.

Reviewer #2

(Remarks to the Author)

In this study, the authors seek to understand metabolic differences in pericytes from patients with PAH. They identify increased glycolysis and activation of de novo pyrimidine metabolic pathways associated with diseased cells. These metabolic changes can be ameliorated somewhat by treatment with CNP, and they dissect signaling mechanisms unique to the PAH cells accounting for different responses to CNP. Overall, the data are good and support their conclusions. I have the following questions and suggestions for the improvement of this manuscript:

1. Can the authors clarify whether they think these changes reflect a different metabolic program that is activated in PAH pericytes or if these metabolic pathways are a consequence of increase growth rates in the cells? In metabolic flux analysis, flux rates are normalized to cell growth to help address this question. Here, ECAR and BrdU incorporation show similar fold changes in Control and PAH cells. Would siCAD attenuate proliferation in control cells to a similar degree as PAH cells? As we contemplate metabolic therapeutics, it is important to consider whether the therapies are more or less targeted to the disease phenotype.

2. It's surprising that CNP treatment was not able to decrease ECAR in control pericytes despite marked down-regulation of HIF1a, GLUT1, and PDK1. The authors suggest this may be related to LDH or MCTs, but these are also targets of HIF1a (MCT4 and LDHA), and should also be associated with decreased glycolysis. It would be helpful to see the effects of these treatments on OCR from the Seahorse data in the Supplement (I'd anticipate an increase in OCR based on these changes) and consider alternative explanations for this discrepancy.

Abstract

- In. 28: Suggest removing the "," after glutaminolysis
- In. 37: "enhanced" mis-spelled

Introduction

- Would mention the decreases in CNP that have been shown in PAH here. Currently mentioned in Discussion, but would also be worth noting up front.

Results

- In. 123: "Seahorse"
- In. 256: I think you mean "¹³C₃-labeled pyruvate". Generally, I would favor referring to labeled metabolites as "M+n metabolite" and referring to the tracer utilized by it's chemical identity (e.g., ¹³C₃-pyruvate).
- I would suggest mentioning in the results that you've previously shown that CNP treatment induces a similar magnitude of cGMP increase in control and PAH cells, suggesting that there are not significant differences in receptor activation to account for the differences that you observe in downstream effects.

Figures

- The methods suggest that isotope data like that in Fig. 1C are presented as the fraction of (M+n) / (M+0) ("all values were normalized to their respective mono-isotopic mass"). Is this the case? Typically these data are presented as (M+n) / sum(M+0, M+1, ..., M+n). The former case suggests only ~ 50% M+6 glucose enrichment in Fig. 1C while the latter suggests ~ 90% enrichment. To be consistent with the field, I would suggest presenting data as (M+n) / sum if it is not presented that way currently.
- I'd caution the authors about conflating isotope enrichment with flux. While I agree that Fig. 1C likely indicates increased glycolytic flux, it could also indicate less enrichment from unlabeled metabolite pools which would give the same result.
- Fig. 2A / Supp. Fig. 2A: I would suggest clarifying the titles to indicate that increased enrichment is associated with PAH cells (e.g., "Enriched in PAH vs. Control").
- It would be helpful to develop a carbon tracing diagram like Fig. 1C for Fig. 2C. Which carbons from glucose and pyruvate are incorporated into which carbons of uracil and cytidine?
- Fig. 3D: The representative immunoblots do not appear to be consistent with the changes demonstrated on the bar graphs.
- Fig. 4B-F: These data should show the mole fraction, which is already normalized, rather than a fraction compared to PDGF-BB treatment. Unclear what fraction of pyruvate is labeled in this experiment, which is very important given multiple sources for unlabeled pyruvate.
- Fig. 4E-F: If there is less labeled pyruvate upstream, there will be less labeled ALA and ASP from the PYR independently of differences in flux. These data should be presented as product / substrate ratios (M+3) ASP % / (M+3) PYR %. I think this analysis would suggest increased flux from PYR to ASP with CNP treatment, not less.

Reviewer #3

(Remarks to the Author)

This is an excellent and well-presented study, supported by a robust dataset that convincingly underpins the authors' conclusions. The findings are highly relevant and contribute meaningfully to our understanding of PAH pathobiology.

I have a few suggestions that I believe could further strengthen the manuscript:

Quantification of Immunofluorescence (Fig. 1g) – Could the authors provide quantification for the immunofluorescence data presented in Fig. 1g? Expanding this analysis to include at least five PAH patients would enhance the statistical rigor and strengthen the conclusions drawn from these findings.

Metabolic Profiling in Different Cell Types – The metabolic characterization of PAH pericytes is highly interesting. Would the authors consider performing similar metabolic measurements in PAH-PAECs and PAH-PASMCs? This would help determine whether the observed metabolic signature is pericyte-specific or shared across vascular cell populations. In particular, assessing de novo pyrimidine synthesis and CAD protein phosphorylation across these cell types could be insightful.

Integration of Acetyl-CoA Metabolism in PAH Pathogenesis – Recent studies (PMC8289565; DOI: 10.1126/scitranslmed.ado7824) have highlighted a significant role of acetyl-CoA metabolism in PAH. Could the authors discuss how their findings align with or differ from these reports? Addressing this connection could provide a more comprehensive view of metabolic dysregulation in PAH.

Use of Human vs. Murine PCLS Models – The use of precision-cut lung slices (PCLS) is commendable; however, given the study's focus on human pericytes, would it not be more relevant to perform these experiments in human lung tissues? Recent studies (DOI: 10.1126/scitranslmed.ado7824; <https://doi.org/10.1161/CIRCULATIONAHA.124.070693>) have demonstrated the feasibility of modeling PAH in control human lungs or using PCLS from PAH patients. Could the authors clarify why murine PCLS were chosen and whether access to human samples is a limiting factor?

Effect of Clinically Available Metabolic Modulators – Given the central role of metabolic dysfunction in PAH, have the authors explored the impact of clinically available metabolic modulators, such as dichloroacetate (DCA), on the identified metabolic signatures? Specifically, does DCA influence pyrimidine synthesis in PAH pericytes?

Broader Metabolic Context in PAH – The discussion could be expanded to integrate the findings within the broader metabolic landscape of PAH. Several metabolic pathways have been implicated in the disease, including the Warburg effect, fatty acid oxidation, pyrimidine synthesis, glutamine/serine metabolism (DOI: 10.1016/j.cmet.2024.04.010), cholesterol and de novo lipid synthesis (DOI: 10.1126/scitranslmed.ado7824), and acetyl-CoA-mediated epigenetic regulation. A more comprehensive discussion reconciling these pathways would strengthen the manuscript's impact and provide a clearer framework for future research directions.

Overall, this is an exciting and well-executed study. Addressing these points would further enhance the manuscript's depth and translational relevance.

Version 1:

Reviewer comments:

Reviewer #1

(Remarks to the Author)

Reviewer #3

(Remarks to the Author)

Nice responses

Answers to Reviewers' comments:

Reviewer #1 (Remarks to the Author):

We would like to thank the reviewer very much for his/her enthusiasm towards our study and the appreciation of the array of orthogonal methods employed. We performed additional analyses and experiments to address all concerns and feel that our manuscript has markedly improved.

Reviewers comments are listed below, followed by our answers (in blue) and an indication where they were addressed in the manuscript (underlined).

#1 Comment 1. There seems to be a systematic problem with the way the statistical analyses were performed that affects many of the figures/panels. It appears that the authors have merged technical and biological replicates, which would artificially inflate statistical power. Since all of these replicates are not 'independent' the two-way ANOVA in this approach would be incorrect. This approach is likely particularly problematic given the modest nature of many of the effect sizes presented.

Response 1. We understand the reviewer's comment and fully acknowledge the concern regarding the merging of technical and biological replicates, which could potentially inflate statistical power and misrepresent significance. In response, we have carefully re-evaluated our statistical analyses and taken the following corrective measures to ensure the robustness and validity of our conclusions:

For all the results, technical replicates were averaged to generate a single value per biological replicate. Statistical analyses (including two-way ANOVA, where applicable) were then performed exclusively on biological replicates, which are independent. This ensures that assumptions of independence underlying parametric tests such as ANOVA are met.

All affected figures have been revised, and statistical revaluations are transparently reported. Only the results in few panels of Figure 2d lose significance, although the previously reported differences between treatment groups are maintained as trend. We have updated the results section to clearly describe the revised statistical approaches accordingly.

Please see results sections (page 6, lines 179 - 184) and methods section (page 38, lines 1089 - 1099) of the revised manuscript. Following figures have been modified:

Figure 1 (a, c, d, h), Figure 2 (d, i), Figure 3 (a, b, c, d), Figure 4 (b, c, d, e, f and h), Figure 5 (b, c, f and g).

Suppl. Figure 3c and Suppl. Figure 8 (a and c).

#1 Comment 2. Key findings (particularly around proliferation) overlap and are almost indistinguishable from their prior manuscript on the antiproliferative effects of CNP. This is not to suggest that this is republishing of prior results, but several of the key findings essentially appear to be a biological replicate experiments of previously published work. This diminishes enthusiasm for the impact of the central finding of CNP antiproliferative effect.

Response 2. We thank the reviewer for their insightful comment. We would like to clarify that only Figure 1a and Figure 3a in the current manuscript represent experiments replicated from our previously published study. This overlap was stated in the text of the results section. However, we agree that this point could be emphasized more clearly, and we have revised the manuscript accordingly.

The Following sentence has been added to the legends of Figures 1a and 3a in the revised manuscript:

'The new experiments illustrated in this panel reproduce findings from our earlier study (Dabral S et al. Commun Biol. 2024) for comparison and validation.'

Also, please see results section describing Fig 1a (page 4 lines 105) and Fig 3a (page 7 lines 237) of the revised manuscript.

For the following reasons we would prefer to keep these Figures in the present manuscript. One of the main objectives of the present study was to investigate the involvement of metabolic reprogramming in the antiproliferative effects of CNP. To properly contextualize these new findings, it was relevant to reproduce selected key experiments from our prior work. Including these replicated data (Figure 1a and Figure 3a) helps to anchor the current study and allows readers—especially those unfamiliar with the earlier publication—to better understand the functional impact of CNP treatment.

Moreover, all other data in this manuscript are novel and expand significantly upon our previous findings by exploring mechanistic links between metabolism and proliferation, which were not addressed before. We hope this clarifies the rationale behind including those specific figures and underscores the added value and originality of the current work.

#1 Comment 3. It is not clear that the 100 nM concentration of CNP used in these studies is 'physiological' or 'pathophysiological'. Given the modest nature of some of the effect sizes seen, this raises the possibility that the findings are largely in vitro artifact.

Response 3. We thank the reviewer for this important observation regarding the concentration of CNP used in our *in vitro* studies. CNP acts predominantly as a paracrine/autocrine factor with rapid clearance from the plasma, primarily through natriuretic peptide clearance receptors (NPR-C) and enzymatic degradation (*Stingo et al. Am J Physiol* **263**, H1318-1321 (1992)). As a result, circulating levels are typically low and unlikely to reflect concentrations within specific tissue microenvironments. In particular, at the endothelial–pericyte interface the local CNP concentrations might be quite high, due to endothelial CNP expression and release. However, precise measurements of endogenous CNP in these niches are lacking.

Given this uncertainty, in consistency with published studies, we selected a concentration of 100 nM CNP for our *in vitro* experiments. As referred below, this concentration has been widely used in functional studies, providing a well-accepted benchmark for assessing biological activity.

1. Kilic A, Rajapurohitam V, Sandberg SM, Zeidan A, Hunter JC, Said Faruq N, Lee CY, Burnett JC Jr, Karmazyn M. A novel chimeric natriuretic peptide reduces cardiomyocyte hypertrophy through the NHE-1-calcineurin pathway. *Cardiovasc Res.* 2010 Dec 1;88(3):434-42.

2. Werner F, Prentki Santos E, Michel K, Schrader H, Völker K, Potapenko T, Krebs L, Abeßer M, Möllmann D, Schlattjan M, Schmidt H, Skryabin BV, Špiranec Spes K, Schuh K, Denton CP, Baba HA, Kuhn M. Ablation of C-type natriuretic peptide/cGMP signaling in fibroblasts exacerbates adverse cardiac remodeling in mice. *JCI Insight*. 2023 Jul 10;8(13):e160416.
3. Day A, Jameson Z, Hyde C, Simbi B, Fowkes R, Lawson C. C-Type Natriuretic Peptide (CNP) Inhibition of Interferon- γ -Mediated Gene Expression in Human Endothelial Cells In Vitro. *Biosensors (Basel)*. 2018 Sep 14;8(3):86.
4. Bachmann JC, Kirchhoff JE, Napolitano JE, Sorota S, Gordon WM, Feric N, Aschar-Sobbi R, Lv J, Cao Z, Coppieters K, Borghetti G, Nyberg M. C-type natriuretic peptide induces inotropic and lusitropic effects in human 3D-engineered cardiac tissue: Implications for the regulation of cardiac function in humans. *Exp Physiol*. 2023 Sep;108(9):1172-1188. doi: 10.1113/EP091303.
5. Mirczuk SM, Lessey AJ, Catterick AR, Perrett RM, Scudder CJ, Read JE, Lipscomb VJ, Niessen SJ, Childs AJ, McArdle CA, McGonnell IM, Fowkes RC. Regulation and Function of C-Type Natriuretic Peptide (CNP) in Gonadotrope-Derived Cell Lines. *Cells*. 2019 Sep 14;8(9):1086.
6. Hirota K, Hirashima T, Horikawa K, Yasoda A, Matsuda M. C-type Natriuretic Peptide-induced PKA Activation Promotes Endochondral Bone Formation in Hypertrophic Chondrocytes. *Endocrinology*. 2022 Mar 1;163(3):bqac005.
7. Perez-Ternero C, Pallier PN, Tremoleda JL, Delogu A, Fernandes C, Michael-Titus AT, Hobbs AJ. C-type natriuretic peptide preserves central neurological function by maintaining blood-brain barrier integrity. *Front Mol Neurosci*. 2022 Oct 4;15:991112.

To support this approach, we have included some of these references in the Methods section of our revised manuscript.

Please see methods section (page 34 lines 947 - 948) of the revised manuscript.

While we acknowledge the inherent limitations of *in vitro* systems, we believe that the consistency of our results across multiple functional assays, together with published studies, support the physiological relevance of our findings. Additionally, the reproduction of key findings in an *ex vivo* model of precision-cut lung slices (Figure 6 of the revised manuscript) further validates our mechanistic conclusions.

#1 Comment 4. More clarity around details of the control and experimental cells would be critical to fully assess the data. There is a Stanford study that is referenced (n=3 control, n=5 PAH) and a German study (controls), which seems to suggest that some of the control cells may have been obtained from an entirely different study? If this is true, there is concern that cells obtained by different investigators and then subjected to different culture programs with variable passage number, may have resulted in metabolic/proliferative changes that are not indicative of the *in vivo* context. There is also variability in the number of PAH and control across figures that doesn't align with the methods. For example, in some instances n=4 controls are reported and n=4 PAH (not the 3 and 5 expected) and therefore what is the rationale for the variability in N and where did the different samples come from across experiments.

Response 4. We thank the reviewer for highlighting the need for greater clarity regarding the source and handling of control and PAH pericytes. We apologize for the ambiguity in the original manuscript.

We initially received pericytes from **3 control** donors (Control Group 1) and **4 PAH** patients from Prof. Vinicio A. de Jesus Pérez (Stanford University). Due to the relatively slow

proliferation of the control pericytes and limited cell numbers, we subsequently **established independent pericyte isolations** in our laboratory in Würzburg, Germany (Control Group 2) to ensure an adequate number of biological replicates for mechanistic experiments. Such pericytes were isolated from healthy lung tissue resected in the course of cancer surgeries. The samples were obtained from our coauthor Prof. Ivan Aleksic.

To ensure consistency between both control groups, all cells were cultured under identical conditions using Pericyte Medium (ScienCell, Cat #1201). Cells from both sources were used at the same passage range (P7–P11). Further, the identity of pericytes isolated in Germany was confirmed by immunostaining for NG2 and PDGFR- β , which matched marker expression in the Stanford-derived cells (Figure R1.a, below). Importantly, both control groups exhibited similar baseline proliferation as well as proliferative responses to PDGF-BB (30 ng/ml, 24 h; Figure R1.b, below). Moreover, as already published in Dabral et al, 2024, both control groups exhibited similar expression levels of the CNP receptor, GC-B (Figure R1.c) and similar cGMP responses to CNP (100 nM) (Figure R1.d).

Figure R1: (a) Representative immunocytochemical stainings of control pericytes from groups 1 (USA) and 2 (Germany) with the pericyte markers PDGFR- β (green) and NG-2 (red). (b - d) Group 1 and Group 2 control pericytes exhibit (b) similar proliferative response to PDGF-BB, (c) GC-B expression and (d) cGMP responses to CNP (100 nM). (b: BrdU incorporation assay, 1-way ANOVA; c: immunoblotting, Student's unpaired t-test; d: RIA, 1-way ANOVA, n = 3 biological replicates from Group 1 and Group 2 control pericytes. Please note that the data in panel c were published before (Dabral et al, Commun Biol. 2024).

Together these data demonstrate that control pericytes derived from the two different sources but cultured under standardized conditions display similar biological behavior. Therefore, to increase statistical power through a higher number of biological replicates we included pericyte samples from both locations in one control group.

The samples were used as follow for the experiments:

- **For metabolomic profiling and target expression analyses** (GLUT-1, phospho-CAD, CAD, PDK1, and PC), we used pericytes **exclusively from Stanford** to maintain group consistency:
 - **Metabolomics:** n = 3 control, n = 3 PAH
 - **Expression studies:** n = 3 control, n = 4 PAH
- **For mechanistic studies**, we used all 4 PAH pericytes, and included the Germany-isolated control pericytes to increase the number of independent biological replicates.

The 'n' numbers are provided in the figure legends of the revised manuscript and the details of the sample usage is provided in the methods section (page 33, lines 929 -934) in the revised manuscript.

#1 Comment 5. It is not entirely surprising that PDGFR stimulation would cause a range of proliferative effects and that this would require uptake of anabolic substrate and similarly that any inhibition of proliferation would at least partially reverse the effects. The CNP effects are interesting, but greater mechanistic clarity would be achieved if there were additional manipulations of what are proposed as the key molecular drivers of the cellular phenotypes:

Response 5 We thank the reviewer for this insightful comment. Our study had two primary objectives: **(1)** to investigate the role of metabolic reprogramming in sustaining the hyperproliferative phenotype of lung pericytes in pulmonary arterial hypertension (PAH), and **(2)** to assess whether the antiproliferative effects of C-type natriuretic peptide (CNP), are mediated, at least in part, through modulation of these metabolic pathways.

As our study focussed on pericyte proliferation (which is known to contribute to microvascular remodeling in PAH), PDGF-BB was chosen as a disease-relevant mitogen due to its strong proliferative effect on pericytes and elevated levels in PAH patients.

Using specific inhibitors and siRNA knockdown experiments, we dissected the third messengers and downstream molecules participating in the antiproliferative and metabolic effects of the CNP/cGMP signaling pathway and found interesting differences between control and PAH pericytes. These are our main observations:

1. The following cyclic GMP-regulated 3rd messengers mediate CNP signaling:

- **cGMP-dependent protein kinase (cGKI): cGKI inhibition** abrogated the effects of CNP on PDGF-BB–induced proliferation (Dabral S *et al.* Commun Biol. 2024) and on the expression of HIF-1 α and GLUT1 (Figure 3g).
- **cGMP-stimulated phosphodiesterase (PDE) 2A: PDE2 inhibition** with BAY 60-7550 mimicked the effects of CNP by reducing PDGF-BB–induced proliferation and CAD phosphorylation (Figures 5c, 5g). Notably, our study also reveals that CNP/cGMP/PDE2A signaling is especially active in PAH pericytes.

2. The following downstream metabolic drivers of proliferation are altered in PAH pericytes and/or inhibited by CNP:

- **GLUT1: GLUT1 inhibition** with BAY-876 reduced both glucose uptake and PDGF-BB–induced hyperproliferation of PAH pericytes (Figures 1h–i), emphasizing the metabolic dependency of the proliferative phenotype.
- **CAD: CAD knockdown in PAH pericytes** markedly diminished PDGF-BB–induced proliferation (Figure 2i), highlighting a reliance on *de novo* pyrimidine synthesis.
- **Notably, in contrast to PAH pericytes, CAD knockdown in control pericytes barely reduced their proliferation** (new data in the revised manuscript). These interesting observations remark the role of CAD in the disease context (Figure 2i of the revised manuscript).

Please see results section (page 7 lines 218 - 226) and corresponding new Figure 2i; and discussion (on page 13, lines 450 - 452).

- **PKD1 inhibition** using dichloroacetate (DCA) (new data, now included) effectively reduced lactate levels, confirming suppression of the Warburg effect. Interestingly, DCA only modestly attenuated PDGF-BB–induced proliferation in PAH pericytes (~18% reduction), and did not significantly affect *de novo* pyrimidine synthesis (DNP_{PyS}) (Suppl. Fig. 4c–g of the revised manuscript). These results suggest that the relatively limited antiproliferative effect of DCA—especially in contrast to the ~75% reduction observed with CNP—may be due to its inability to inhibit DNP_{PyS}. Since DNP_{PyS} is a key metabolic pathway supporting nucleotide biosynthesis and cell proliferation in PAH pericytes, its continued activity may allow for sustained proliferation despite partial metabolic modulation by DCA.

Please see results section (page 6-7, line 204 - 212) Suppl. Figure 4 c - g in the revised manuscript and discussion (page 14, lines 481 – 487; page 14, lines 502 - 503) in the revised manuscript.

These results dissect molecular differences between control and PAH pericytes. Moreover, they demonstrate that CNP, through cGMP signaling, inhibits the hyperproliferation and metabolic remodeling of PAH pericytes by targeting GLUT1 and CAD. Our research concentrated on these pathways but does not exclude the involvement of additional cGMP-modulated antiproliferative pathways.

Reviewer #2 (Remarks to the Author):

We would like to thank the reviewer very much for his/her interest in our study, the careful readings and all comments and suggestions. They were indeed very helpful. We performed additional analyses and experiments to address all concerns and feel that our manuscript has markedly improved.

Reviewers comments are listed below, followed by our answers (in blue) and an indication where they were addressed in the manuscript (underlined).

#2 Comment 1. Can the authors clarify whether they think these changes reflect a different metabolic program that is activated in PAH pericytes or if these metabolic pathways are a

consequence of increase growth rates in the cells? In metabolic flux analysis, flux rates are normalized to cell growth to help address this question.

Response 1. Thank you for this insightful question. To address whether the observed metabolic changes (CAD phosphorylation/activation) represent a distinct program in PAH pericytes or simply are a consequence of increased proliferation, we performed comparative analyses in another hyperproliferative cell type, namely in cultured lung fibroblasts from patients with Idiopathic pulmonary fibrosis (IPF).

As shown in Figure R2.a (below), cultured IPF fibroblasts showed enhanced proliferation in the presence of PDGF-BB (50 ng/ml, 24 h), in comparison to control fibroblasts. CNP (100 nM) significantly attenuated PDGF-BB induced proliferation of both control and IPF fibroblasts, similar to our findings in lung pericytes. Notably, at difference to the pericytes, this proliferative responses of fibroblasts to PDGF-BB and their attenuation by CNP were not accompanied by changes in CAD phosphorylation or expression (Figure R 2.b). Furthermore, CAD phosphorylation, indicative of its enzymatic activity, was either unchanged when normalized to β -actin or even reduced when normalized to total CAD levels in IPF fibroblasts compared to controls (Figure R2.c).

Figure R2 (for reviewer purposes only): (a) CNP (100 nM) prevents PDGF-BB (50 ng/ml)-induced proliferation of human control and IPF lung fibroblasts as analysed by BrdU incorporation, n = 4 fibroblasts from controls and IPF lungs, 2-way ANOVA, *p<0.05 vs PBS, #p<0.05 vs PDGF-BB, \$ p<0.05 vs PDGF-BB-control. (b) PDGF-BB (50ng/ml) and CNP (100nM) did not alter CAD phosphorylation and CAD expression in control and IPF fibroblasts. n = 4 controls and IPF fibroblasts, 1-way ANOVA. (c) CAD phosphorylation and CAD expression are similar in controls and IPF fibroblasts. n =6 controls and 8 IPF fibroblasts, Unpaired student's t-test.

These findings indicate that increased proliferation alone is not sufficient to induce CAD phosphorylation, supporting our conclusion that PAH pericytes engage a distinct metabolic program that involves specific activation of the CAD-driven pyrimidine biosynthesis pathway.

We prefer to omit this figure in the manuscript, because part of these data will be included in a separate study on lung fibrosis. Moreover, the present manuscript is very long already.

Secondly, we agree that metabolic flux analysis, especially when normalized to cell growth, is a more definitive method to distinguish intrinsic metabolic changes from proliferation-driven effects. In our original submission, we inadvertently referred to our ¹³C-isotope enrichment data as “flux” causing confusion. This has been corrected throughout, and we now describe these studies appropriately as isotope enrichment studies. Our conclusions are based on observed differences in pathway engagement, not on quantitative flux rates.

Our single time-point enrichment analysis was designed to capture steady-state labeling patterns that indicate relative pathway engagement. While this approach lacks the temporal resolution of true flux analysis, it is widely used to infer pathway activity, particularly when supported by functional assays and targeted interventions.

The alignment of our isotope enrichment data, with changes observed in carbamoyl aspartate levels, CAD phosphorylation in human patients' lungs and the selective impact of CAD knockdown in PAH pericytes support our conclusion that these cells exhibit a distinct metabolic program. This suggests that activation of *de novo* pyrimidine synthesis is not simply a byproduct of proliferation but reflects a disease-specific metabolic dysregulation.

Finally, we do acknowledge the limitation of single time-point analysis and hence, have clearly added it as limitation to our study.

For study limitations, please see discussion section (page 16, lines 569 - 574).

#2 Comment 2. Here, ECAR and BrdU incorporation show similar fold changes in Control and PAH cells. Would siCAD attenuate proliferation in control cells to a similar degree as PAH cells? As we contemplate metabolic therapeutics, it is important to consider whether the therapies are more or less targeted to the disease phenotype.

Response 2. Based on the reviewer's suggestion, we performed additional experiments to directly compare the effects of CAD knockdown on proliferation of control and PAH pericytes under PDGF-BB stimulation. In control pericytes, PDGF-BB increased proliferation by approximately 1.8-fold, and CAD knockdown only mildly reduced this proliferative response. In contrast, PAH pericytes showed a markedly enhanced proliferation under PDGF-BB stimulation, which was significantly attenuated by siCAD (by approximately 50%).

These findings suggest that CAD plays a prominent role in sustaining the hyperproliferative phenotype of PAH pericytes, supporting the idea that CAD inhibition preferentially targets a disease-specific metabolic dysregulation. This indicates the translational relevance of CAD as a potential therapeutic target in PAH.

Please see results section (page 7 lines 218 - 226) and corresponding new Figure 2i; and discussion (on page 13, lines 450 - 452).

#2 Comment 3. It's surprising that CNP treatment was not able to decrease ECAR in control pericytes despite marked down-regulation of HIF1a, GLUT1, and PDK1. The authors suggest this may be related to LDH or MCTs, but these are also targets of HIF1a (MCT4 and LDHA), and should also be associated with decreased glycolysis. It would be helpful to see the effects of these treatments on OCR from the Seahorse data in the Supplement (I'd anticipate an increase in OCR based on these changes) and consider alternative explanations for this discrepancy.

Response 3: As suggested by the reviewer, we have added the OCR data from the seahorse assay to the supplement. As seen in Suppl. Fig 6b of the revised manuscript, OCR remained unchanged following both PDGF-BB stimulation and CNP treatment in both control and PAH pericytes, under the tested conditions.

Interestingly, although CNP treatment significantly reduced ECAR and intracellular pyruvate levels in PAH pericytes, no such changes were observed in control cells, despite comparable downregulation of HIF1 α , GLUT1, and PDK1. This selective metabolic response may be due to intrinsic differences in pyruvate kinase isoform expression. Specifically, PAH pericytes exhibited a significantly higher PKM2/PKM1 ratio compared to controls, primarily due to lower PKM1 expression (Suppl. Figure 5a–b). A high PKM2/PKM1 ratio is known to promote aerobic glycolysis (Warburg effect) as seen in cancer cells (Tamada M *et al. Clin Cancer Res.* 2012 Oct 15;18(20):5554-61). Our observation is also consistent with the findings of Zhang et al., who reported reduced PKM1 expression and enhanced PKM2-driven glycolysis in pulmonary adventitial fibroblasts from PAH patients (Zhang H *et al. Circulation.* 2017 Dec 19;136(25):2468-2485).

These differences may render PAH pericytes more susceptible to glycolytic modulation by CNP. In contrast, control pericytes with a lower PKM2/PKM1 ratio may rely less on glycolysis and exhibit limited responsiveness in ECAR to CNP, even when upstream HIF1 α targets are downregulated.

Based on these findings, we speculate that CNP may modulate PKM isoform expression or activity in PAH pericytes, leading to reduced glycolytic flux and pyruvate/lactate levels. Future studies will investigate this potential regulatory axis in more detail.

Please see results section (page 8 lines 267 - 275, 282 - 283) and corresponding new Suppl. Figure 5 and Suppl. Figure 6b.

#2 Comment 4. Abstract

- In. 28: Suggest removing the "," after glutaminolysis
- In. 37: "enhanced" mis-spelled

Response 4: We thank the reviewer for his/her very careful reading. These mistakes were corrected.

#2 Comment 5. Introduction

- Would mention the decreases in CNP that have been shown in PAH here. Currently mentioned in Discussion, but would also be worth noting up front.

Response 5. The decreases in CNP levels in experimental PH and clinical PAH are mentioned in introduction in the revised version.

Please see introduction section (page 3 - 4 lines 76 - 78).

#2 Comment 6. Results

- In. 123: "Seahorse"

Response 6. The mistake has been corrected.

#2 Comment 7. Results- In. 256: I think you mean "¹³C₃-labeled pyruvate". Generally, I would favor referring to labeled metabolites as "M+n metabolite" and referring to the tracer utilized by its chemical identity (e.g., ¹³C₃-pyruvate).

Response 7. We thank for the opportunity to clarify. We did not use ¹³C₃-pyruvate as a tracer. Hence, the pyruvate mentioned in the mentioned line is the labelled metabolite: Pyruvate (M + 3). To avoid confusions, we have now referred all the labelled metabolites as 'Metabolite (M + n)' and only the tracer is described by its chemical identity as ¹³C₆-glucose.

#2 Comment 8. Results - I would suggest mentioning in the results that you've previously shown that CNP treatment induces a similar magnitude of cGMP increase in control and PAH cells, suggesting that there are not significant differences in receptor activation to account for the differences that you observe in downstream effects.

Response 8. Thank you for the suggestion. We have added the text to the results section.

Please see introduction section (page 2 line 73) and results section (page 10 lines 330 - 332).

#2 Comment 9. Figures

- The methods suggest that isotope data like that in Figure 1C are presented as the fraction of (M+n) / (M+0) ("all values were normalized to their respective mono-isotopic mass"). Is this the case? Typically these data are presented as (M+n) / sum(M+0, M+1, ..., M+n). The former case suggests only ~ 50% M+6 glucose enrichment in Figure 1C while the latter suggests ~ 90% enrichment. To be consistent with the field, I would suggest presenting data as (M+n) / sum if it is not presented that way currently.

Response 9. Thank you for the comment. We appreciate the opportunity to clarify. The isotope labeling data in Figure 1C (and throughout the manuscript) are presented as the normalized mass isotopomer distribution, calculated as (M+n) / sum(M+0 to M+n) for each metabolite. We agree that this is the standard approach in the field and ensures accurate representation of isotopic enrichment.

In Figure 4b, 4c, 4e and 4f, these normalized isotopomer distribution are represented as fold-change vs PDGF-BB.

We have revised the figure legends and the methods section to make this normalization approach explicitly clear and avoid any confusion.

Please see methods section (page 35 lines 989 - 995) of the revised manuscript.

#2 Comment 10. Figures - I'd caution the authors about conflating isotope enrichment with flux. While I agree that Figure 1C likely indicates increased glycolytic flux, it could also indicate less enrichment from unlabelled metabolite pools which would give the same result.

Response 10. We thank the reviewer for this important clarification. We agree that isotope enrichment alone does not directly equate to metabolic flux, as enrichment patterns can also be influenced by unlabelled metabolite pools and other factors. In our study, we have presented isotope enrichment data. For clarity, we have revised the relevant sections of the text to clearly distinguish enrichment from flux inference. The interpretation of Figure 1C has also been updated to reflect this distinction.

Please see Figure legends 1c and Suppl. Figure 3b and results section (page 4, lines 108-109, 120, 122, and page 8, line 265, 284)

#2 Comment 11. - Figure 2A / Suppl. Figure 2A: I would suggest clarifying the titles to indicate that increased enrichment is associated with PAH cells (e.g., "Enriched in PAH vs. Control").

Response 11. The titles have been modified accordingly.

#2 Comment 12. - It would be helpful to develop a carbon tracing diagram like Figure 1C for Figure 2C. Which carbons from glucose and pyruvate are incorporated into which carbons of uracil and cytidine?

Response 12. The carbon tracing program has been added for Figure 2c of the revised manuscript.

#2 Comment 13. The representative immunoblots do not appear to be consistent with the changes demonstrated on the bar graphs.

Response 13. The blots have been replaced (this does not change the data, which were derived from the evaluation of several independent western blots).

#2 Comment 14. These data should show the mole fraction, which is already normalized, rather than a fraction compared to PDGF-BB treatment. Unclear what fraction of pyruvate is labeled in this experiment, which is very important given multiple sources for unlabeled pyruvate.

Response 14. We thank the reviewer for this insightful comment. While our main figures present labeling data normalized to PDGF-BB-treated samples to emphasize relative group differences and minimize variability arising from biological sample heterogeneity, we agree that mole fractions provide valuable context for understanding absolute labeling.

To address this, we have included the corresponding mole fraction data of pyruvate labeling in the supplementary data (Suppl. Figure 6a). This allows for direct interpretation of labeling patterns, including the contribution of unlabeled pyruvate, while maintaining a focus in the main figures on the treatment-dependent changes we aimed to highlight.

We hope this dual presentation satisfies the reviewer's concern.

Please see Suppl. Figure 6a and c of the revised manuscript.

#2 Comment 15. - Figure 4E-F: If there is less labeled pyruvate upstream, there will be less labeled ALA and ASP from the PYR independently of differences in flux. These data should be presented as product / substrate ratios (M+3) ASP % / (M+3) PYR %. I think this analysis would suggest increased flux from PYR to ASP with CNP treatment, not less.

Response 15. We thank the reviewer for the suggestion to present enrichment data as product/substrate ratios (e.g., aspartate M+3 / pyruvate M+3). As requested, we performed this analysis (Figure R3). While informative, the resulting ratios remain largely unchanged—or slightly increased, as the reviewer anticipated—and do not yield additional biological insight beyond what is already conveyed by the absolute enrichment data.

Figure R3: Product/Substrate ratios for Lactate M + 3, Alanine M + 3 and Aspartate M + 3 divided by Pyruvate M + 3.

The reduction in aspartate M+3 might be partly derived from decreased pyruvate availability, as evidenced by a reduction in ECAR (reflecting diminished glycolytic activity and lactate production) and a modest decrease in alanine M+3 (Figure 4d–e). However, the more pronounced decrease in aspartate M+3 suggests a selective rerouting of pyruvate metabolism away from aspartate biosynthesis, rather than a uniform downregulation of all pyruvate-derived outputs (see scheme in 4a).

This selective shift is further supported by a reduction in total carbamoyl aspartate levels, indicating downstream consequences on *de novo* pyrimidine synthesis (Figure 4h). Together, these findings indicate that CNP affects DNPys through two mechanisms: by reducing pyruvate availability and by modulating CAD phosphorylation, a key regulatory event in the pathway. The effect on pyruvate levels may also be influenced by CNP-mediated changes in pyruvate kinase isoform expression (see Response to Comment #2.3).

We believe that presenting absolute isotopologue enrichment, in conjunction with functional and metabolic data, more accurately reflects the biological impact of CNP in this context.

Finally, we would like to emphasize that the metabolomic data represent single time-point isotope enrichment, which is indicative of pathway engagement under specific conditions, but does not allow a quantitative metabolic flux analysis. We have clearly added this as limitation to our study.

For limitations, please see discussion section (page 16 lines 569 - 574).

Reviewer #3 (Remarks to the Author):

We would like to thank the reviewer very much for his/her interest in our study. His/her comment that “our findings are highly relevant and contribute meaningfully to our understanding of PAH pathobiology” was very motivating for us. We performed additional experiments to address his/her concerns and feel that our manuscript has markedly improved.

Reviewers comments are listed below, followed by our answers (in blue) and an indication where they were addressed in the manuscript (underlined).

#3 Comment 1. Quantification of Immunofluorescence (Figure 1g) – Could the authors provide quantification for the immunofluorescence data presented in Figure 1g? Expanding this analysis to include at least five PAH patients would enhance the statistical rigor and strengthen the conclusions drawn from these findings.

Response 1. We thank the reviewer for this valuable suggestion. In response, we have now performed quantification of the immunofluorescence data and included the results the revised manuscript.

Regarding sample size, while we fully agree that increasing the number of PAH patient samples would enhance statistical power, we were limited by sample availability. Nonetheless, we have increased the number of biological replicates from $n = 3$ to $n = 4$, which improves the robustness of our analysis while remaining within the constraints of the available clinical material. We believe the updated quantification supports our original observations and conclusions.

Please see revised Figure 1g and Suppl. Figure 1b for Glut-1, Figure 2g and Suppl. Figure 2 for pCAD_{Thr456} and Suppl. figure 4b for PDK-1.

#3 Comment 2: Metabolic Profiling in Different Cell Types – The metabolic characterization of PAH pericytes is highly interesting. Would the authors consider performing similar metabolic measurements in PAH-PAECs and PAH-PASMCs? This would help determine whether the observed metabolic signature is pericyte-specific or shared across vascular cell populations. In particular, assessing de novo pyrimidine synthesis and CAD protein phosphorylation across these cell types could be insightful.

Response 2. We thank the reviewer for the suggestion. Indeed, metabolic characterization of PAH- pulmonary artery endothelial cells (PAECs) and PAH- pulmonary artery smooth muscle cells (PASMCs) is interesting and necessary. Such studies underline the importance of metabolic reprogramming in vascular dysfunction. While we did not directly assess PAECs or PASMCs in the current study, several studies have already characterized metabolic reprogramming in these cell types. Hernandez-Saavedra et al. performed stable isotope

tracing in PH-derived PAECs and PASMCs, revealing distinct metabolic adaptations: PASMCs displayed increased glycolysis and elevated pentose phosphate pathway (PPP) flux, which was further enhanced by TGF- β treatment. In contrast, PAECs exhibited increased levels of glycolytic intermediates, reduced PPP flux, and enhanced glutamine-derived anaplerosis with decreased fatty acid oxidation (Hernandez-Saavedra D et al. Sci Rep. 2020 Jan 15;10(1):413). Similarly, Fessel et al. used steady-state metabolomics to study PAECs expressing mutant BMPR2—a common heritable cause of PAH—and found elevated glycolytic intermediates, suppressed β -oxidation, and decreased glutamine utilization (Fessel JP et al. Pulm Circ. 2012 Apr-Jun;2(2):201-13).

Moreover, PH adventitial fibroblasts have also been shown to exhibit significant metabolic rewiring, including increased glucose uptake, accumulation of proximal glycolytic intermediates, and elevated pyruvate and lactate (Li M et al. Circulation. 2016 Oct 11;134(15):1105-1121). These cells also displayed changes in redox homeostasis, as reflected by increased levels of both reduced and oxidized glutathione, along with evidence of enhanced NADPH production via the PPP (Li M et al. Circulation. 2016 Oct 11;134(15):1105-1121).

Together, these studies demonstrate that metabolic remodelling is a hallmark of multiple vascular cell types in PAH. In our study, we chose to focus specifically on pericytes, which remain underexplored despite their emerging relevance in pulmonary vascular remodelling. While increased glycolysis (Warburg effect) appears to be a common feature across multiple vascular cell types in PAH, including pericytes, PAECs, PASMCs, and fibroblasts, our here reported novel findings remark the impact of distinct, cell type-specific metabolic adaptations. For example, differences in glutamine metabolism, fatty acid oxidation, and pentose phosphate pathway activity have been reported between PAECs and PASMCs (Hernandez-Saavedra D et al. Sci Rep. 2020 Jan 15;10(1):413). Our novel data add an important piece of information showing that pericytes from PAH lungs exhibit a distinct metabolic program characterized by selective activation of *de novo* pyrimidine synthesis, as evidenced by CAD phosphorylation. Importantly, this phenotype was not observed in other hyperproliferative cells, such as lung fibroblasts from Idiopathic pulmonary fibrosis (IPF) patients, suggesting cell type specificity (**please see our comment 1 to reviewer 2 and Figure R2**). This highlights the importance of considering both shared and unique metabolic programs of different vascular cell populations when studying the metabolic mechanisms of pulmonary vascular remodelling.

Please see discussion section (page 12, lines 427 - 431) in the revised manuscript.

#3 Comment 3: Integration of Acetyl-CoA Metabolism in PAH Pathogenesis – Recent studies (PMC8289565; DOI: 10.1126/scitranslmed.ado7824) have highlighted a significant role of acetyl-CoA metabolism in PAH. Could the authors discuss how their findings align with or differ from these reports? Addressing this connection could provide a more comprehensive view of metabolic dysregulation in PAH.

Response 3. We thank the reviewer for the comment. Pulmonary arterial hypertension (PAH) involves a metabolic shift in vascular cells toward glycolysis and anaplerosis, bypassing mitochondrial oxidative phosphorylation. This reprogramming supports rapid energy production and supplies precursors for proliferation and vascular remodelling (Valuparampil Varghese M et al. J Clin Med. 2020 Feb 6;9(2):443, Shi J et al. Am J Physiol Heart Circ Physiol. 2020 Sep 1;319(3):H613-H631, Peng H et al. Front Cell Dev Biol. 2021 Feb 18;9:626047). Normally, glycolysis-derived pyruvate enters mitochondria for conversion to acetyl-CoA and

TCA cycle fueling (Martínez-Reyes I et al. Nat Commun. 2020 Jan 3;11(1):102). In PAH, however, pyruvate is increasingly diverted to biosynthetic pathways due to high proliferative demand or mitochondrial suppression. Our findings demonstrate that proliferating PAH pericytes preferentially utilize aerobic glycolysis and divert pyruvate toward aspartate synthesis, which feeds into the *de novo* pyrimidine synthesis pathway. This process is catalyzed by CAD, a rate-limiting enzyme whose expression and activating phosphorylation are elevated in PAH pericytes. Notably, CAD inhibition significantly reduces PAH pericyte proliferation, underscoring its role in pathological pericyte dysfunction. These results align with recent study, which identify acetyl-CoA metabolism as a central driver of PAH pathogenesis (Grobs Y et al. Sci Transl Med. 2024 Dec 11;16(777)). ATP-citrate lyase (ACLY), a key enzyme that cleaves cytosolic citrate into acetyl-CoA and oxaloacetate, is upregulated in pulmonary arteries of PAH patients (Grobs Y et al. Sci Transl Med. 2024 Dec 11;16(777)). ACLY-derived cytosolic acetyl-CoA supports lipid biosynthesis for membrane formation, while oxaloacetate can be converted into aspartate, feeding nucleotide synthesis via CAD (Lin R et al. Mol Cell. 2013 Aug 22;51(4):506-518, Sullivan LB et al. Cell. 2015 Jul 30;162(3):552-63, Li G et al. Int J Mol Sci. 2021 Sep 23;22(19):10253). Thus, ACLY activity may enhance both lipid and nucleotide biosynthesis, supporting proliferative and remodelling phenotypes.

Our data complement these findings by providing a mechanistic link between pyruvate metabolism, CAD activity, and nucleotide biosynthesis in PAH pericytes. Together, these insights highlight a broader metabolic network—integrating acetyl-CoA metabolism, fatty acid synthesis, and pyrimidine biosynthesis—that sustains pathological cell proliferation in PAH. Further investigation is warranted to delineate the interplay between these interconnected pathways.

Please see discussion section (page 13 lines 459 - 476) in the revised manuscript.

#3 Comment 4: Use of Human vs. Murine PCLS Models – The use of precision-cut lung slices (PCLS) is commendable; however, given the study's focus on human pericytes, would it not be more relevant to perform these experiments in human lung tissues? Recent studies (DOI: 10.1126/scitranslmed.ado7824; <https://doi.org/10.1161/CIRCULATIONAHA.124.070693>) have demonstrated the feasibility of remodelling PAH in control human lungs or using PCLS from PAH patients. Could the authors clarify why murine PCLS were chosen and whether access to human samples is a limiting factor?

Response 4. We thank the reviewer for this thoughtful comment. We fully agree that human PCLS would provide an important and clinically relevant platform, particularly in the context of human pericyte biology. At the time of our study, however, human lung tissues suitable for generating PCLS were not available, and the PCLS technique using human samples had not yet been established in our laboratory. As a result, we utilized murine PCLS, which offered a practical and well-characterized alternative for exploring our mechanistic questions within an intact tissue context.

We acknowledge the translational value of human PCLS, and we are currently working to standardize this technique in our lab. However, this will not be possible within the time frame for the revision of our manuscript. We therefore hope the reviewer will consider the murine PCLS data acceptable in the current context.

#3 Comment 5: Effect of Clinically Available Metabolic Modulators – Given the central role of metabolic dysfunction in PAH, have the authors explored the impact of clinically available

metabolic modulators, such as dichloroacetate (DCA), on the identified metabolic signatures? Specifically, does DCA influence pyrimidine synthesis in PAH pericytes?

Response 5. Thank you for a very interesting question. Dichloroacetate, inhibitor of Pyruvate dehydrogenase Kinase (PDK) is a well-established and clinically viable metabolic modulator. By inhibiting PDK, DCA routes pyruvate into the TCA cycle, reducing lactate production. Previous studies have shown that DCA enhances glucose oxidation and mitigates excessive proliferation and apoptosis resistance of PSMCs in vitro (Li, B. et al. *Int J Mol Med* 42, 1391-1400 (2018). Furthermore, DCA has been shown to attenuate PAH in experimental animal models and improve hemodynamics and exercise capacity in patients with idiopathic PAH in a 4-month Phase 1 clinical trial (Michelakis ED et al. *Sci Transl Med.* 2017 Oct 25;9(413), Michelakis ED et al. *Circulation.* 2002 Jan 15;105(2):244-50, McMurtry MS et al. *Circ Res.* 2004 Oct 15;95(8):830-4).

Following this interesting (and interested!) comment of the reviewer and considering that PDK1 expression was increased in PAH pericytes (Suppl. Figure 4a), we investigated whether DCA affects the metabolic signatures identified in PAH pericytes, specifically DNP_{PyS}. We performed steady-state metabolic profiling using [U-¹³C₆] glucose isotope tracing in PDGF-BB-treated PAH pericytes with or without DCA (10 mM, 24 h) treatment. As expected, DCA significantly reduced lactate levels, confirming effective inhibition of the Warburg effect (Suppl. Figure 4c). DCA treatment also led to a modest but statistically significant reduction (~18%) in PDGF-BB-induced pericyte proliferation (Suppl. Figure 4d). Interestingly, DCA had no detectable effect on aspartate M+3 labeling or carbamoyl aspartate levels (Suppl. Figure 4e - f). Furthermore, no reduction was observed in M + 5 and M + 2 labelled nucleotides (indicative of active DNP_{PyS}). This suggests that DNP_{PyS} was not affected by DCA (Suppl. Figure 4g). No effect on DNP_{PyS} is likely due to the rerouting of pyruvate toward acetyl-CoA synthesis, which may subsequently fuel the TCA cycle, support fatty acid biosynthesis, or contribute to epigenetic modifications. Importantly, the relatively mild antiproliferative effect of DCA, especially when compared to CNP (~75% reduction) may be attributed to its lack of inhibition on DNP_{PyS}. Since DNP_{PyS} is a key metabolic pathway supporting nucleotide production and cell proliferation in PAH pericytes, its continued activity may sustain pericyte proliferation despite the partial metabolic reprogramming induced by DCA.

Please see results section (page 6-7, line 205 - 214) Suppl. Figure 4 c - g in the revised manuscript and discussion (page 14, lines 484 – 490; page 15 - 16, lines 505 - 507) in the revised manuscript.

#3 Comment 6: Broader Metabolic Context in PAH – The discussion could be expanded to integrate the findings within the broader metabolic landscape of PAH. Several metabolic pathways have been implicated in the disease, including the Warburg effect, fatty acid oxidation, pyrimidine synthesis, glutamine/serine metabolism (DOI: 10.1016/j.cmet.2024.04.010), cholesterol and de novo lipid synthesis (DOI: 10.1126/scitranslmed.ado7824), and acetyl-CoA-mediated epigenetic regulation. A more comprehensive discussion reconciling these pathways would strengthen the manuscript's impact and provide a clearer framework for future research directions.

Response 6. We thank the reviewer for this important suggestion. We agree that contextualizing our findings within the broader metabolic framework of PAH significantly strengthens the manuscript.

PAH is increasingly recognized as a disease of metabolic dysregulation, involving multiple intersecting pathways. These include the Warburg effect/aerobic glycolysis (Culley MK and Chan SY. *J Clin Invest.* 2018 Aug 31;128(9):3704-3715.), suppression of fatty acid oxidation (Lee MH et al. *Am J Physiol Lung Cell Mol Physiol.* 2022 Sep 1;323(3):L355-L371), upregulation of glutamine and serine metabolism (Rachedi NS et al. *Cell Metab.* 2024 Jun 4;36(6):1335-1350.e8), dysregulated lipid and cholesterol synthesis (Grobs Y *et al.* *Sci Transl Med.* 2024 Dec 11;16(777)), and acetyl-CoA-mediated epigenetic changes. Our study contributes to this growing body of work by identifying *de novo* pyrimidine synthesis (DNP_{PyS})—specifically via CAD activation—as a key metabolic driver of pericyte proliferation in PAH. Importantly, our data show that PDGF-BB-treated PAH pericytes increase pyruvate accumulation and CAD activity, linking glycolytic flux directly to nucleotide biosynthesis. This integrates with previous reports, such as those by Ma et al., demonstrating a role for *de novo* purine synthesis via ATIC in driving smooth muscle cell proliferation in PAH. Similarly, work by Zhao et al. identified ATP-citrate lyase (ACLY) as a metabolic hub that generates acetyl-CoA for lipid synthesis and oxaloacetate for aspartate production—feeding into DNP_{PyS} through CAD. These findings converge on a central theme: PAH vascular cells co-opt central carbon metabolism to fuel both membrane and nucleotide biosynthesis, enabling pathological remodeling.

Although our study did not directly assess glutamine or serine metabolism, these pathways are metabolically interconnected with glycolysis and nucleotide synthesis. Glutamine provides nitrogen and carbon for nucleotide biosynthesis (Zhang J et al. *EMBO J.* 2017 May 15;36(10):1302-1315.), while serine contributes one-carbon units for purine/pyrimidine synthesis and NADPH for redox homeostasis (Pan S et al. *Int J Oncol.* 2021 Feb;58(2):158-170.). Both depend on glycolytic intermediates—3-phosphoglycerate and glucose-derived carbon backbones—which are upregulated in PAH pericytes.

Our findings therefore extend this metabolic framework by linking pyruvate metabolism, CAD activation, and DNP_{PyS} to a broader anabolic and regulatory network that includes lipid synthesis, amino acid metabolism, redox balance, and epigenetic control. Further investigation is warranted to delineate the interplay between these interconnected pathways.

The discussion in the revised manuscript was extended to include these studies. *Please see discussion section (page 13 - 14 lines 467 – 479) of the revised manuscript.*

Answers to Reviewers' comments:

Reviewer #1 (Remarks to the Author):

We would like to thank the reviewer very much for acknowledging our efforts with the revision and the appreciation for the work.

Reviewers comments are listed below, followed by our answers (in blue) and an indication where they were addressed in the manuscript (underlined).

#1 Comment 1. Their explanation of problems expanding the control primary human cell lines reinforces one of my concerns around the data in PAH pericytes relative to control: namely that the control and experimental cells were derived from different centers with different protocols. At the very least, a great attention in the text to this as a limitation would be indicated in my opinion.

Response 1. We understand the concern of the reviewer. We have added the following statement to the methods section in the revised manuscript:

'As the control pericytes were derived from two different centers, which may introduce variability; we acknowledge this as a limitation when interpreting group comparisons.'

Please see methods section (page 17 lines 604 - 605) of the revised manuscript.

Reviewer #3 (Remarks to the Author):

We would like to thank the reviewer very much for his/her interest in our study and for appreciation of our efforts.